# Ecosystem-level Analysis of Deployed Machine Learning Reveals Homogeneous Outcomes

**Connor Toups**[*]
Stanford University

**Rishi Bommasani**[*][†]
Stanford University

**Kathleen A. Creel**
Northeastern University

**Sarah H. Bana**
Chapman University

**Dan Jurafsky**
Stanford University

**Percy Liang**
Stanford University

## Abstract

Machine learning is traditionally studied at the model level: researchers measure and improve the accuracy, robustness, bias, efficiency, and other dimensions of specific models. In practice, however, the societal impact of any machine learning model depends on the context into which it is deployed. To capture this, we introduce *ecosystem-level analysis*: rather than analyzing a single model, we consider the collection of models that are deployed in a given context. For example, ecosystem-level analysis in hiring recognizes that a job candidate's outcomes are determined not only by a single hiring algorithm or firm but instead by the collective decisions of all the firms to which the candidate applied. Across three modalities (text, images, speech) and eleven datasets, we establish a clear trend: deployed machine learning is prone to *systemic failure*, meaning some users are exclusively misclassified by all models available. Even when individual models improve over time, we find these improvements rarely reduce the prevalence of systemic failure. Instead, the benefits of these improvements predominantly accrue to individuals who are already correctly classified by other models. In light of these trends, we analyze medical imaging for dermatology, a setting where the costs of systemic failure are especially high. While traditional analyses reveal that both models and humans exhibit racial performance disparities, ecosystem-level analysis reveals new forms of racial disparity in model predictions that do not present in human predictions. These examples demonstrate that ecosystem-level analysis has unique strengths in characterizing the societal impact of machine learning.[1]

## 1   Introduction

Machine learning (ML) is pervasively deployed. Systems based on ML mediate our communication and healthcare, influence where we shop or what we eat, and allocate opportunities like loans and jobs. Research on the societal impact of ML typically focuses on the behavior of individual models. If we center people, however, we recognize that the impact of ML on our lives depends on the aggregate result of our many interactions with ML models.

In this work, we introduce *ecosystem-level analysis* to better characterize the societal impact of machine learning on people. Our insight is that when a ML model is deployed, the impact on users depends not only on its behavior but also on the behavior of other models and decision-makers (left of Figure 1). For example, the decision of a single hiring algorithm to reject or accept a candidate does

---

[*]Equal contribution.

[†]Corresponding author: `nlprishi@stanford.edu`.

[1]All code is available at `https://github.com/rishibommasani/EcosystemLevelAnalysis`.

37th Conference on Neural Information Processing Systems (NeurIPS 2023).

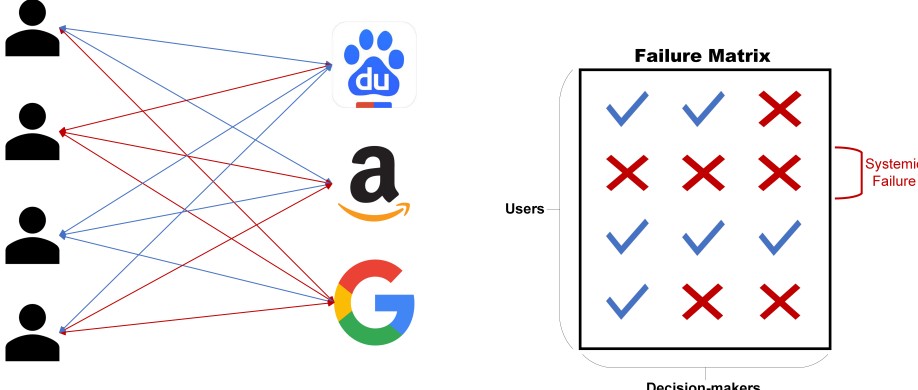

Figure 1: **Ecosystem-level analysis.** Individuals interact with decision-makers (*left*), receiving outcomes that constitute the failure matrix (*right*).

not determine whether or not the candidate secures a job; the outcome of her search depends on the decisions made by all the firms to which she applied. Likewise, in selecting consumer products like voice assistants, users choose from options such as Amazon Alexa, Apple Siri, or Google Assistant. From the user's perspective, what is important is that at least one product works.

In both settings, there is a significant difference from the user's perspective between *systemic failure*, in which *zero* systems correctly evaluate them or work for them, and any other state. The difference in marginal utility between zero acceptances and one acceptance is typically much higher than the difference between one acceptance and two acceptances. This non-linearity is not captured by average error metrics. For example, imagine that three companies are hiring and there are ten great candidates. All three companies wrongly reject the same two great candidates for a false negative error rate of 20%. Now imagine that each company wrongly rejects *different* candidates. The second decision ecosystem has the same false negative error rate, 20%, but no systemic failures or jobless candidates.

Ecosystem-level analysis is a methodology that centers on the *failure matrix* ($F$; right of Figure 1): $F$ encapsulates the outcomes individuals receive from all decision-makers. Of special interest are *systemic failures*: exclusively negative outcomes for individuals such as misclassifications or rejections from all decision-makers [Bommasani et al., 2022]. To establish general trends in deployed machine learning, we draw upon a large-scale audit [**HAPI**; Chen et al., 2022a] that spans three modalities (images, speech, text), three commercial systems per modality, and eleven datasets overall. Because **HAPI** contains predictions from some of the largest commercial ML providers – including Google, Microsoft, and Amazon – and the models evaluated are deployed models that real users interact with, evaluating **HAPI** has real-world implications.

Across all settings, ecosystem-level analysis reveals a consistent pattern of *homogeneous outcomes*. In each of the **HAPI** datasets, many instances are classified correctly by all three commercial systems and many instances are classified incorrectly by all three systems. The pattern of outcomes across the decision ecosystem is *homogenous* if the rates of systemic failure and consistent classification success significantly exceed the rate predicted by independent instance-level behavior. Since the commercial systems analyzed in this dataset are popular and widely used, being failed by all systems in the dataset has societally meaningful consequences.

Ecosystem-level analysis enriches our understanding not only of the status quo, but also of how models change over time. In particular, it allows us to ask, when individual models improve, how do ecosystem-level outcomes change? Since **HAPI** tracks the performance of the same systems over a three-year period, we consider all cases where at least one of the commercial systems improves. For example, Amazon's sentiment analysis API reduced its error rate on the WAIMAI dataset by 2.5% from 2020 to 2021; however, this improvement did not decrease the systemic failure rate at all. Precisely 0 out of the model's 303 improvements are on instances on which all other models had failed. These findings generalize across all cases: on average, just 10% of the instance-level improvement of a single commercial system occurs on instances misclassified by all other models. This is true even though systemic failures account for 27% of the instances on which the models *could have* improved. Thus most model improvements do not significantly reduce systemic failures.

To build on these trends, we study medical imaging, a setting chosen because the costs to individuals of systemic failure of medical imaging classification are especially high. We compare outcomes from prominent dermatology models and board-certified dermatologists on the **DDI** dataset [Daneshjou et al., 2022]: both models and humans demonstrate homogeneous outcomes, though human outcomes are more homogenous. Given established racial disparities in medicine for both models and humans, fairness analyses in prior work show that both humans and models consistently perform worse for darker skin tones (e.g. Daneshjou et al. [2022] show lower ROC-AUC on **DDI**). Ecosystem-level analysis surfaces new forms of racial disparity in models that do not present in humans: models are more *homogenous* when evaluating images with darker skin tones, meaning that all systems agree in their correct or incorrect classification, whereas human homogeneity is consistent across skin tones.

Our work contributes to a growing literature on the *homogeneous outcomes* of modern machine learning [Ajunwa, 2019, Engler, 2021, Creel and Hellman, 2022, Bommasani et al., 2022, Fishman and Hancox-Li, 2022, Wang and Russakovsky, 2023, Jain et al., 2023]. While prior work conceptualizes these phenomena, our work introduces new methodology to study these problems and provides concrete findings for a range of ML deployments spanning natural language processing, computer vision, speech, and medical imaging. Further, by centering individuals, we complement established group-centric methods [Barocas and Selbst, 2016, Buolamwini and Gebru, 2018, Koenecke et al., 2020], unveiling new forms of racial disparity. Ecosystem-level analysis builds on this existing work, providing a new tool that contributes to holistic evaluations of the societal impact of machine learning.

Developing better methodologies for ecosystem-level analysis of deployed machine learning systems is important for two reasons. First, systemic failures in socially consequential domains could exclude people from accessing goods such as jobs, welfare benefits, or correct diagnoses. Individuals who are failed by only one model can gain informal redress by switching to another model, for example by seeking second doctor's opinion or switching banks. Individuals failed by *all* models cannot. Socially consequential systemic failures can happen due to reliance on APIs, such as image recognition APIs used to identify cancers, speech recognition APIs used to verify individuals for banking, or facial recognition APIs used to unlock devices. Systemic failures can also occur in algorithmic decision-making systems such as those used for hiring, lending, and criminal justice. The social importance of avoiding systemic failures in all of these systems is clear.

Second, as decision-makers become more likely to rely on the same or similar algorithms to make decisions [Kleinberg and Raghavan, 2021] or to use the same or similar components in building their decision pipelines [Bommasani et al., 2022], we believe that the prevalence of systemic failures could increase. Measuring systemic failures as they arise with the tools we present in this paper will expand our understanding of their prevalence and likely causes.

## 2 Ecosystem-level Analysis

How individuals interact with deployed machine learning models determines ML's impact on their lives. In some contexts, individuals routinely interact with multiple ML models. For example, when a candidate applies to jobs, they typically apply to several firms. The decision each company makes to accept or reject the candidate may be mediated by a single hiring algorithm. In other contexts, individuals select a single model from a set of options. For example, when a consumer purchases a voice assistant, they typically choose between several options (e.g. Amazon Alexa, Google Assistant, Apple Siri) to purchase a single product (e.g. Amazon Alexa). Centering people reveals a simple but critical insight: exclusively receiving negative outcomes, as when individuals are rejected from every job or unable to use every voice assistant, has more severe consequences than receiving even one positive outcome.

### 2.1 Definition

Recognizing how ML is deployed, we introduce *ecosystem-level analysis* as a methodology for characterizing ML's cumulative impacts on individuals. Consider $N$ individuals that do, or could, interact with $k$ decision-makers that apply $\hat{y}$ labels according to their decision-making processes $h_1, \ldots, h_k$. Individual $i$ is associated with input $x_i$, label $y_i$, and receives the label $\hat{y}_i^j = h_j(x_i)$ from decision-maker $j$.

**Outcomes.** Define the *failure matrix* $F \in \{0,1\}^{N \times k}$ such that $F[i,j] = \mathbb{I}\left[\hat{y}_i^j \neq y_i\right]$. The *failure outcome profiles* $\mathbf{f}_i$ for individual $i$, which we refer to as the outcome profile for brevity, denotes $F[i,:]$. The *failure rate* $\bar{f}_j$ for decision-maker $j$ is $\bar{f}_j = \frac{\sum_{i=1}^N F[i,j]}{N}$ (i.e. the empirical classification error in classification). For consistency, we order the entries of decision-makers (and, thereby, the columns of the failure matrix) in order of ascending failure rate: $F[:,1]$ is the outcome profile associated with the decision-maker with the fewest failures and $F[:,k]$ is the outcome profile associated with the decision-maker with the most failures. The failure matrix is the central object in ecosystem-level analysis (see Figure 1).

**Systemic Failures.** Individual $i$ experiences *systemic failure* if they exclusively experience failure across the domain of interest: $F[i,:] = [1,\ldots,1]$. Not only are systemic failures the worst possible outcomes, but they also often result in additional harms. If an individual applying to jobs is rejected everywhere, they may be unemployed. If no commercial voice assistant can recognize an individual's voice, they may be fully locked out of accessing a class of technology. In our ecosystem-level analysis, we focus on systemic failures as a consequential subset of the broader class of *homogeneous outcomes* [Bommasani et al., 2022].

## 3 Homogeneous Outcomes in Commercial ML APIs (HAPI)

To establish general trends made visible through ecosystem-level analysis, we draw upon a large-scale three-year audit of commercial ML APIs [**HAPI**; Chen et al., 2022a] to study the behavior of deployed ML systems across three modalities, eleven datasets, and nine commercial systems.

### 3.1 Data

Chen et al. [2022a] audit commercial ML APIs, tracking predictions across these APIs when evaluated on the same eleven standard datasets over a period of three years (2020 – 2022). We consider ML APIs spanning three modalities (text, images, speech), where each modality is associated with a task (SA: sentiment analysis, FER: facial emotion recognition, SCR: spoken command recognition) and 3 APIs per modality (e.g. IBM, Google, Microsoft for spoken command recognition). The models evaluated are from Google (SA, SCR, FER), Microsoft (SCR, FER), Amazon (SA), IBM (SCR), Baidu (SA), and Face++ (FER). Additionally, each modality is associated with three to four datasets, amounting to eleven datasets total; further details are deferred to the supplement.

To situate our notation, consider the DIGIT dataset for spoken command recognition and the associated APIs (IBM, Google, Microsoft). For each instance (i.e. image) $x_i$ in DIGIT, the outcome profile $\mathbf{f}_i \in \{0,1\}^3$ is the vector of outcomes. The entries are ordered by ascending model failure rate: $F[:,1]$ corresponds to the most accurate model (Microsoft) and $F[:,3]$ corresponds to the least accurate model (Google).

| | Facial emotion recognition | | | | Spoken command recognition | | | Sentiment analysis | | | |
| --- | --- | --- | --- | --- | --- | --- | --- | --- | --- | --- | --- |
| | RAFDB | AFNET | EXPW | FER+ | FLUENT | DIGIT | AMNIST | SHOP | YELP | IMDB | WAIMAI |
| **Dataset size** | 15.3k | 287.4k | 31.5k | 6.4k | 30.0k | 2.0k | 30.0k | 62.8k | 20.0k | 25.0k | 12.0k |
| **Number of classes** | 7 | 7 | 7 | 7 | 31 | 10 | 10 | 2 | 2 | 2 | 2 |
| $h_1$ **failure rate** (i.e. error) | 0.283 | 0.277 | 0.272 | 0.156 | 0.019 | 0.217 | 0.015 | 0.078 | 0.043 | 0.136 | 0.110 |
| $h_2$ **failure rate** (i.e. error) | 0.343 | 0.317 | 0.348 | 0.316 | 0.025 | 0.259 | 0.015 | 0.095 | 0.111 | 0.219 | 0.151 |
| $h_3$ **failure rate** (i.e. error) | 0.388 | 0.359 | 0.378 | 0.323 | 0.081 | 0.472 | 0.043 | 0.122 | 0.486 | 0.484 | 0.181 |
| **Systemic failure rate** | 0.152 | 0.178 | 0.181 | 0.066 | 0.01 | 0.129 | 0.002 | 0.039 | 0.021 | 0.043 | 0.065 |

Table 1: **Basic statistics on HAPI datasets** including the (observed) systemic failure rate (i.e. fraction of instances misclassified by all models).

**Descriptive statistics.** To build general understanding of model performance in **HAPI**, we provide basic descriptive statistics (Table 1). For most datasets, all APIs achieve accuracies within 5–10% of each other (exceptions include DIGIT, YELP, IMDB). Interestingly, we often find the systemic failure rate is roughly half the failure rate of the most accurate model $h_1$.

## 3.2 Ecosystem-level Behavior

In order for a measure of systemic failure to be useful, it must be (i) meaningful and (ii) comparable across systems. A challenge to the meaningfulness of any proposed metric is that systemic failures occur more often in an ecosystem with many inaccurate models. A metric for systemic failure that primarily communicated the aggregate error rates of models in the ecosystem would not be meaningful as an independent metric. It also would not support goal (ii) because we could not compare the rates of systemic failure across ecosystems with varying model accuracies. It would be difficult to identify a system with a 'large' rate of systemic failure because the systemic failure properties would be swamped by the error rates of the models in the ecosystem. Therefore, achieving meaningfulness and comparability requires the metric to incorporate error correction.

Assuming model independence is a helpful baseline because it adjusts for model error rates without making assumptions about the correlation between models in the ecosystem. To avoid assumptions and for the sake of simplicity, therefore, we juxtapose the *observed* behavior with a simple theoretical model in which we assume models fail independently of each other. Under this assumption, the distribution of the *baseline* number of model failures $t \in \{0, \dots, k\}$ follows a Poisson-Binomial distribution parameterized by their failure rates (Equation 2).

The baseline of independence also means that our metric does not attempt to quantify whether it is "reasonable" that the models all fail on some instances. For example, some instances might be harder (or easier) than others, making it more likely that all models will fail (or succeed) to classify that instance. However, "hardness" is observer-relative. What is hard for one class of models might be easy for another class of model, and likewise with humans with particular capabilities or training. Therefore 'correcting' the metric to account for hardness would relativize the metric to the group of humans or class of models for whom that instance is hard. We choose independence as a baseline to be neutral on this point. However, we depart from independence in Appendix §A.3, exploring how a baseline that assumes some level of correlation between models can more accurately model the observed distribution of ecosystem-level outcomes.

Comparing the true observed distribution of ecosystem-level outcomes with the baseline distribution helps illuminate how correlated outcomes are across models. Below we define $P_{\text{observed}}$ (Equation 1) and $P_{\text{baseline}}$ (Equation 2).

$$P_{\text{observed}}(t \text{ failures}) = \frac{\sum_{i=1}^{N} \mathbb{I}\left[t = \sum_{j=1}^{k} F[i,j]\right]}{N} \tag{1}$$

$$P_{\text{baseline}}(t \text{ failures}) = \text{Poisson-Binomial}(\bar{f}_1, \dots, \bar{f}_k)[t] \tag{2}$$

**Finding 1: Homogenous Outcomes.**   In Figure 2a, we compare the observed and baseline distributions for the spoken command recognition dataset DIGIT. We find the observed ecosystem-level outcomes are more clearly *homogenous* compared to the baseline distribution: the fraction of instances that receive either extreme outcome (all right or all wrong) exceeds the baseline rate. These findings generalize to all the datasets (Figure 2b): the observed rate always exceeds the baseline rate for the homogeneous outcomes (above the line $y = x$) and the reverse mostly holds for intermediary outcomes.

## 4   Do Model Improvements Improve Systemic Failures?

The performance of a deployed machine learning system changes over time. Developers serve new versions to improve performance [Chen et al., 2022b,a], the test distribution shifts over time [Koh et al., 2021], and the users (sometimes strategically) change their behavior [Björkegren et al., 2020]. In spite of this reality, most analyses of the societal impact of machine learning only consider static models.

Ecosystem-level analysis provides a new means for understanding how models change and how those changes impact people. When models change, what are the broader consequences across their model ecosystem? Do single-model improvements on average improve ecosystem-level outcomes by reducing systemic failures? And to what extent are the individuals for whom the model improves the same individuals were previously systemically failed?

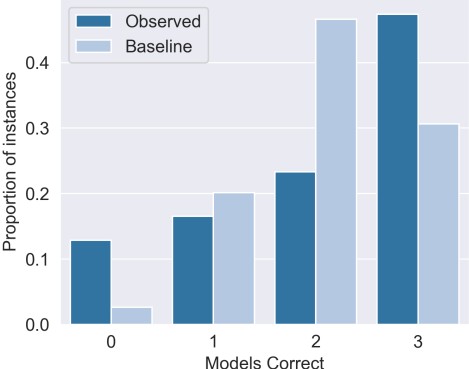

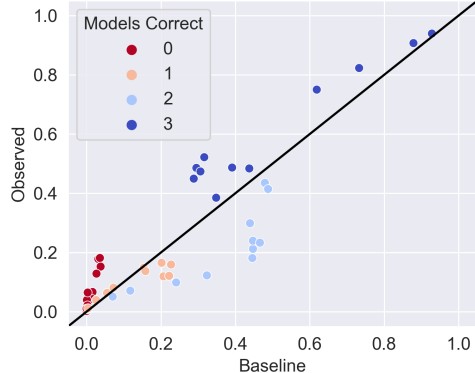

(a) Ecosystem-level outcomes for DIGIT dataset.  (b) Ecosystem-level outcomes for all datasets.

Figure 2: **Homogeneous outcomes.** Ecosystem-level analysis surfaces the general trend of *homogeneous outcomes*: the observed rates that all models succeed/fail consistently exceeds the corresponding baseline rates. Figure 2a shows that models in the DIGIT dataset are more likely to all fail or all succeed on an instance than baseline. Figure 2b shows that across all datasets, systemic failure (red dots) and consistent success (blue dots) of all three models on an instance are both more common than baseline, whereas intermediate results are less common than baseline.

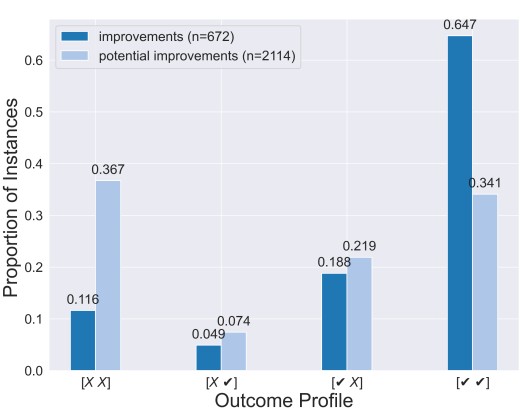

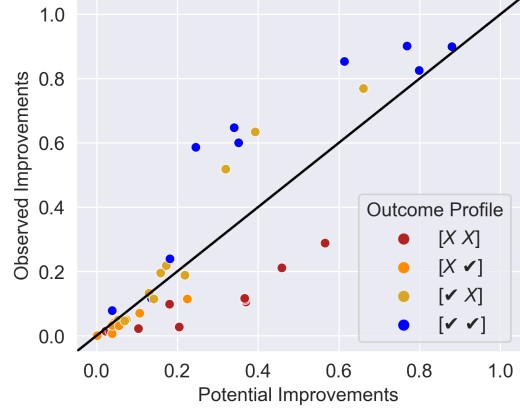

(a) Outcome profile distribution for (Baidu, Google) when Amazon improves on WAIMAI from 2020 to 2021.

(b) The distribution of outcome profiles for all year-over-year model improvements across all datasets.

Figure 3: **Model improvement is not concentrated on systemic failures.** When a model improves, we compare the distribution of outcome profiles of the other two models on its initial failures (*potential improvements*) to the distribution on the instances it improved on (*observed improvements*). Across all improvements, including Amazon's improvement on WAIMAI (*left*), there is a clear over-improvement on [✓, ✓] (above $y = x$ on *right*) and under-improvement on [X, X] (below the identity line on *right*).

**Setup.** Chen et al. [2022a] evaluated the performance of the commercial APIs on the same eleven evaluation datasets each year in 2020–2022. Of all year-over-year comparisons, we restrict our attention to cases where one of the three APIs for a given task improves by at least 0.5% accuracy.[2] Let $h_{imp}$ denote the model that improved. We identify the instances that $h_{imp}$ initially misclassified in the first year as *potential improvements* and the subset of these instances that $h_{imp}$ correctly classified in the second year as *improvements*. Considering the initial distribution of failures for $h_{imp}$, we can ask where does the $h_{imp}$ improve? We answer this by comparing the distribution of outcome profiles for the *other* models (besides $h_{imp}$) between the potential improvement and improvement sets.

---

[2]The supplement §B.3 contains an analysis that confirms our findings are robust to alternate thresholds.

**Finding 2: Model improvements make little progress on systemic failures.** As a case study, we consider Amazon's improvement on the WAIMAI dataset from 2020 to 2021. In Figure 3a, from left to right, [X, X] indicates the other APIs (Baidu, Google) both fail, [X, ✓] and [✓, X] indicate exactly one of the other APIs succeed, and [✓, ✓] indicates both succeed. The majority (64.7%) of Amazon's improvement is on instances already classified correctly by the other two APIs, far exceeding the fraction of potential improvements that were classified correctly by Baidu and Google (34.1%) in 2020. In contrast, for systemic failures, the improvement of 11.6% falls far short of the potential improvement of 36.7%. In fact, since models can also fail on instances they previously classified correctly, the model's improvement on systemic failures is even worse in terms of net improvement. Amazon's improvement amounts to *no net reduction of systemic failures*: the model improves on 78 systemic failures but also regresses on 78 instances that become new systemic failures, amounting to no net improvement. The Baidu and Google APIs similarly show little improvement on systemic failures even as models improve.

This finding is not unique to Amazon's improvement on the WAIMAI dataset: in the 11 datasets we study, we observe the same pattern of homogeneous improvements from 2020-2022. In Figure 3b, we compare the observed improvement distribution[3] ($y$ axis) to the potential improvement distributions ($x$ axis) across all model improvements. We find a clear pattern: systemic failures (the [X, X] category) are represented less often in the observed improvement set than in the potential improvement set. This finding indicates that when models improve, they *under-improve* on users that are already being failed by other models. Instead, model improvements especially concentrate on instances where both other models succeeded already.

Ecosystem-level analysis in the context of model improvements disambiguates two plausible situations that are otherwise conflated: does single-model improvement (i) marginally or (ii) substantively reduce systemic failures? We find the reduction of systemic failures is consistently marginal: in every case, the reduction fails to match the the distribution seen in the previous year (i.e. every [X, X] red point is below the line in Figure 3b).

## 5 Ecosystem-level Analysis in Dermatology (DDI)

Having demonstrated that ecosystem-level analysis reveals homogeneous outcomes across machine learning deployments, we apply the ecosystem-level methodology to medical imaging. We consider this setting to be an important use of ecosystem-level analysis because machine learning makes predictions that inform the high-stakes treatment decisions made by dermatologists.

### 5.1 Data

Daneshjou et al. [2022] introduced the Diverse Dermatology Images (**DDI**) dataset of 656 skin lesion images to evaluate binary classification performance at detecting whether a lesion was malignant or benign. Images were labelled with the ground-truth based on confirmation from an external medical procedure (biopsy). In addition, each image is annotated with skintone metadata using the Fitzpatrick scale according to one of three categories: Fitzpatrick I & II (light skin), Fitzpatrick III & IV (medium skin), and Fitzpatrick V & VI (dark skin). We use the predictions from Daneshjou et al. [2022] on **DDI** for two prominent ML models (ModelDerm [Han et al., 2020] and DeepDerm [Esteva et al., 2017]) and two board-certified dermatologists.[4] We defer further details to Appendix C.

### 5.2 Results

**Finding 3: Both humans and models yield homogeneous outcomes; humans are more homogeneous.** We compare *observed* and *baseline* ecosystem-level outcomes on **DDI** for models (Figure 4a) and humans (Figure 4b). Consistent with the general trends in **HAPI**, model predictions yield homogeneous outcomes. For human predictions from board-certified dermatologists, we also

---

[3]The supplement §B.4 contains analysis comparing 'net improvements' to 'potential improvements' as well; the trends are consistent across both analyses.

[4]Daneshjou et al. [2022] also evaluated a third model [HAM10K; Tschandl et al., 2018] that almost always predicts the majority class in this class-imbalanced setting. We exclude this model since its failures are not interesting, but replicate our analyses in the supplement Appendix C to show the findings still hold if it is included.

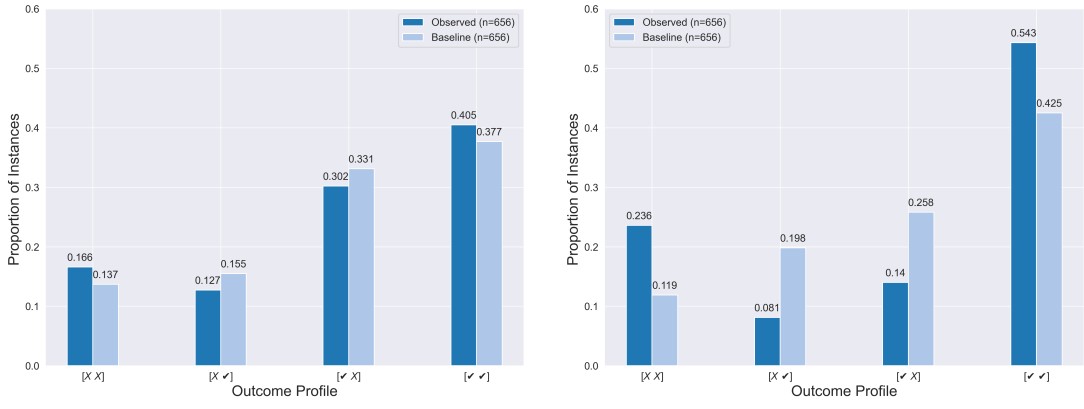

(a) Outcome profiles for models.   (b) Outcome profiles for humans.

Figure 4: **Homogeneous outcomes for models and humans.** Consistent with **HAPI**, model predictions (*left*) yield homogenous outcomes on **DDI**. Human predictions (*right*) are even more homogenous than models.

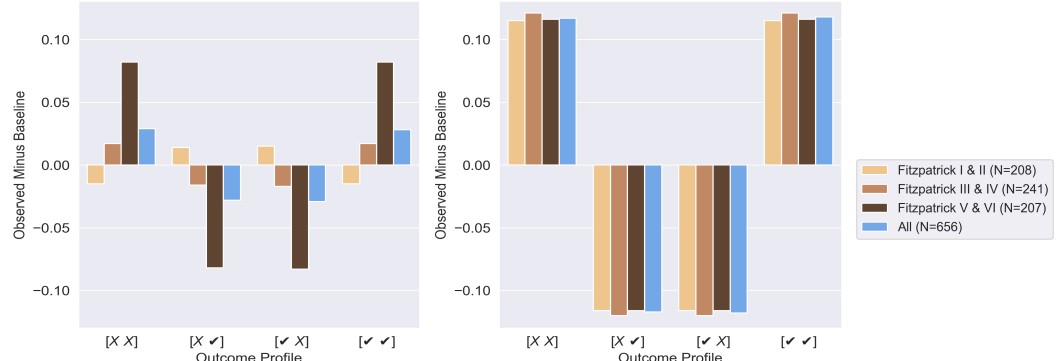

(a) Model outcome profiles by skin tone.   (b) Human outcome profiles by skin tone.

Figure 5: **Racial disparities for models but not humans.** We stratify ecosystem-level analysis in **DDI** by the three skin tone categories, plotting the difference between *observed* and *baseline* rates. Models (*left*) show clear racial disparities, exhibiting the most homogeneity for the darkest skin tones, whereas humans (*right*) show no significant racial disparity.

see that outcomes are homogeneous. However, in comparing the two, we find that humans yield even more homogeneous outcomes. We take this to be an important reminder: while we predict that models are likely to produce homogeneous outcomes, we should also expect humans to produce homogeneous outcomes, and in some cases more homogeneous outcomes.

**Finding 4: Ecosystem-level analysis reveals new racial disparities in models but not humans.** Standard analyses of machine learning's impact focus on model performance across groups [e.g. Buolamwini and Gebru, 2018, Koenecke et al., 2020]. In AI for medicine, several works have taken such an approach towards understanding fairness [Obermeyer et al., 2019, Seyyed-Kalantari et al., 2021, Kim et al., 2022, Colwell et al., 2022]. Daneshjou et al. [2022] demonstrate racial disparities for both model and human predictions on **DDI** and find that predictive performance is worst for darkest skin tones, aligning with broader trends of racial discrimination in medicine and healthcare [Vyas et al., 2020, Williams and Wyatt, 2015, Fiscella and Sanders, 2016].

Ecosystem-level analysis can build on these efforts. We conduct the same analysis from Figure 4 stratified by skin tone. In previous experiments, we had observed systemic failures across the whole population. Here we measure systemic failures for subpopulations grouped by skin tone. In Figure 5, we plot the difference between the observed and baseline rates on the $y$ axis: the **All** bars (*blue*) reproduce the homogeneity results from Figure 4.

Strikingly, ecosystem-level-analysis surfaces an important contrast between model behavior (*left*) and human behavior (*right*). Models (Figure 5a) are most homogenous for darkest skin tones (Fitzpatrick V & VI; *dark brown*) and least homogeneous for the lightest skin tones (Fitzpatrick I & II; *cream*): The observed systemic failure rate for the darkest skin tones is 8.2% higher than the baseline, while for the lightest skin tones it is 1.5% lower than the baseline. By contrast, humans (Figure 5b) show no significant variation as a function of skin tone.

Ecosystem-level analysis therefore identifies a new form of racial disparity not previously documented. Critically, while prior works show racial disparities for both models and humans, here we find a form of racial disparity that is salient for models but not present for humans. We note that, in absolute terms, homogeneity is higher for all racial groups in human predictions than model predictions, though human predictions don't display significant differences in homogeneity across racial group. This demonstrates that ecosystem-level analysis can reveal new dimensions of fairness, allowing stakeholders to identify the metrics that are most relevant in their context, be that model error or systemic failure rate. Our tool helps researchers and stakeholders evaluate the holistic ecosystem of algorithmic – and human – judgements that ultimately shapes outcomes for those subject to algorithmic judgement.

## 6 Commentary

While ecosystem-level analysis reveals new dimensions of machine learning's societal impact, it also opens the door for further questions. We prioritize two here, deferring further discussion and a longer related work section to the supplement. How can we *explain* the pervasive homogeneous outcomes we have observed in deployed machine learning? And what are the *implications* of this work for both researchers and policymakers?

### 6.1 Explanations for Homogeneous Outcomes

Our findings provide evidence across several settings for homogeneous outcomes in deployed machine learning (Finding 1; §3) that are mostly unabated by model improvement (Finding 2; §4).

**Data-centric explanations.** We posit that "example difficulty" may give rise to homogeneous outcomes and provide three analyses that instantiate variants of this hypothesis.

First, *human ambiguity* on the ground-truth may predict homogeneous outcomes. To test this, we make use of the ten human annotations per example in the FER+ dataset within **HAPI**. We find that the systemic failure rate is monotonically correlated with annotator disagreement with the majority label. This suggests that some systemic failures are correlated with the ambiguity or "hardness" of the image. However, we find that even some of the least ambiguous images, namely images on which all or most annotators agree, have systemic failures. . This indicates that human ambiguity is only partially explanatory of ecosystem-level model outcomes. We explore this further in §A.1.

Second, *human error* may predict homogeneous outcomes. To test this, we compare human predictions on **DDI** with the ground truth biopsy results. We stratify ecosystem-level analysis by dermatologist performance, comparing (i) instances both dermatologists get right, (ii) instances they both get wrong, and (iii) instances where they disagree and exactly one is right. We find that when both dermatologists fail, there continues to be outcome homogenization. However, when both dermatologists succeed, there is no homogeneity and the observed rates almost exactly match the baseline rates for every image. This suggests human error is also partially predictive of ecosystem-level model outcomes. We explore this further in §A.2.

Finally, more *expressive theoretical models* can better capture the observed trends than our simple full instance-level independence model. We introduce a two-parameter model. $\alpha$ fraction of instances are categorized as difficult and the remaining $1 - \alpha$ are easy. A model's failure rate $\bar{f}_j$ over all examples scales to $(1 + \Delta)\bar{f}_j$ on hard examples and $\left(1 - \frac{\alpha\Delta}{1-\alpha}\right)\bar{f}_j$ on easy examples. Partitioning examples in this way, while continuing to assume instance-level independence, inherently homogenizes: models are more likely to systemically succeed on easy instances and systemically fail on hard instances. To best fit the **HAPI** data, the resulting average $\alpha$ $(0.2 - 0.3)$ and $\Delta$ $(1 - 4,$ meaning these examples are $2 - 5\times$ harder) values are quite extreme. In other words, for this theoretical model to fit the

data, a significant fraction ($\approx 25\%$) would need to be considerably harder ($\approx 3.5\times$) than the overall performance. We explore this further in §A.3.

These analyses contribute to an overarching hypothesis that example *difficulty* partially explains homogeneous outcomes. While we discuss the construct of difficulty in the supplement, we draw attention to two points. First, difficulty is largely in the eye of the beholder: what humans or models perceive as difficult can differ [e.g. adversarial examples; Goodfellow et al., 2014]. Thus while example difficulty could be caused by inherent properties of the example (such as noisy speech or corrupted images), it could just as well be due to model properties, such as all the models having made similar architectural assumptions or having parallel limitations in their training data. Second, whether or not homogeneous outcomes are caused by example difficulty does not change their societal impact. The consequences of systemic failure can be material and serious (e.g. unemployment).

**Model-centric Explanations and Algorithmic Monoculture.** An alternative family of explanations put forth in several works is that correlated outcomes occur when different deployed machine learning models share training data, model architectures, model components, learning objectives or broader methodologies [Ajunwa, 2019, Engler, 2021, Bommasani et al., 2021, Creel and Hellman, 2022, Fishman and Hancox-Li, 2022, Bommasani et al., 2022, Wang and Russakovsky, 2023]. Such *algorithmic monoculture* [Kleinberg and Raghavan, 2021, Bommasani et al., 2022] in fact appears to be increasingly common as many deployed systems, including for the tasks in **HAPI**, are derived from the same few foundation models [Bommasani et al., 2023, Mądry, 2023]. Unfortunately, we are unable to test these hypotheses because we know very little about these deployed commercial systems, but we expect they form part of the explanation and encourage future work in this direction.

**Implications for Researchers.** Our paper shows that ecosystem-level research can reveal previously-invisible social impacts of machine learning. We believe this methodology concretizes the impact of decision-making in real contexts in which individuals would typically be affected by decisions from many actors (e.g. job applications, medical treatment, loan applications, or rent pricing). As we demonstrate in **DDI**, our methodology applies equally to human-only, machine-only, and more complex intertwined decision-making. We anticipate understanding of homogeneous outcomes arising from any or all of these sources will be valuable.

**Implications for Policymakers.** Given the pervasive and persisting homogeneous outcomes we document, there may be a need for policy intervention. In many cases no single decision-maker can observe the decisions made by others in the ecosystem, so individual decision-makers (such as companies) may not know that systemic failures exist. In addition, systemic failures are not currently the responsibility of any single decision-maker, so no decision-maker is incentivized to act alone. Consequently, policy could implement mechanisms to better monitor ecosystem-level outcomes and incentivize ecosystem-level improvement. In parallel, regulators should establish mechanisms for recourse or redress for those currently systemically failed by machine learning [see Voigt and von dem Bussche, 2017, Cen and Raghavan, 2023].

## 7  Conclusion

We introduce ecosystem-level analysis as a new methodology for understanding the cumulative societal impact of machine learning on individuals. Our analysis on **HAPI** establishes general trends towards homogeneous outcomes that are largely unaddressed even when models improve. Our analysis on **DDI** exposes new forms of racial disparity in medical imaging that arise in model predictions but not in human predictions. Moving forward, we hope that future research will build on these empirical findings by providing greater theoretical backing, deeper causal explanations, and satisfactory sociotechnical mitigations. To ensure machine learning advances the public interest, we should use approaches like ecosystem-level analysis to holistically characterize its impact.

## Acknowledgments and Disclosure of Funding

We would like to thank Shibani Santurkar, Mina Lee, Deb Raji, Meena Jagadeesan, Judy Shen, p-lambda, and the Stanford ML group for their feedback on this work. We would like to thank James Zou and Lingjiao Chen for guidance with using the **HAPI** dataset. We would like to thank Roxana Daneshjou for providing the **DDI** dataset along with guidance on how to analyze the dataset. In addition, the authors would like to thank the Stanford Center for Research on Foundation Models (CRFM) and Institute for Human-Centered Artificial Intelligence (HAI) for providing the ideal home for conducting this interdisciplinary research. RB was supported by the NSF Graduate Research Fellowship Program under grant number DGE-1655618. This work was supported in part by a Stanford HAI/Microsoft Azure cloud credit grant and in part by the AI2050 program at Schmidt Futures (Grant G-22-63429).

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

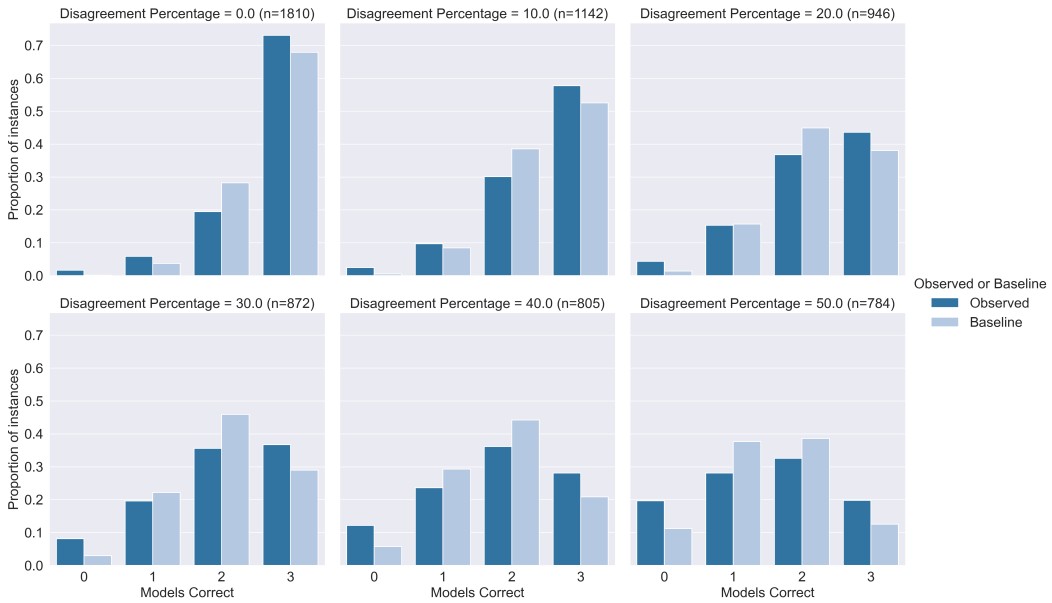

Figure 6: We stratify ecosystem-level analysis on FER+ by instance-level annotator disagreement – which we take as a proxy for the ambiguity inherent to the input instance – and plot the difference between *observed* and *baseline* rates for each instance subset. We observe homogeneous outcomes for all subsets of data, regardless of the level of annotator disagreement on the instance.

## A  Data-centric Explanations for Homogeneous Outcomes

Prior work has explored model-centric explanations for homogeneous outcomes [Ajunwa, 2019, Engler, 2021, Bommasani et al., 2021, Creel and Hellman, 2022, Fishman and Hancox-Li, 2022, Bommasani et al., 2022, Wang and Russakovsky, 2023]. However, data-centric explanations are comparatively less explored. We posit that properties of the underlying data could contribute to the homogeneous outcomes that we observe in all of the datasets we examine. Intuitively, if we believe that some examples are 'hard' and others are 'easy', then we might expect to see models all fail for the 'hard' examples and all succeed for the 'easy' examples.

We test three variants of this hypothesis. In §A.1, we examine how the level of annotator disagreement in the ground truth label impacts ecosystem-level behavior. In §A.2, we test how the accuracy of human dermatologists in predicting the malignancy of a skin lesion image correlates with homogeneous outcomes. Finally, to build on these finer-grained empirical analyses, in §A.3, we introduce a more express theoretical model. Under this model, parameterized by two difficulty parameters, we compute a different baseline rate for ecosystem-level outcomes, showing it can better recover the observation distribution.

### A.1  Annotator disagreement

To study the effects of annotator disagreement, we make use of the FER+ dataset. Each instance of the FER+ dataset, a facial emotion recognition dataset, contains emotion annotations from 10 human annotators; the emotion label is determined by majority vote [Barsoum et al., 2016]. Because each instance has been annotated by multiple annotators, we can calculate the annotator disagreement for each instance and use this as a proxy for the 'ambiguity' of the instance. For example, an instance where all 10 annotators agree that the label is 'sad' is less ambiguous than an instance where 6 annotators vote the label should be 'fear' and 4 vote that the label should be 'surprise'.

The test set of FER+ provided in **HAPI** contains instances with disagreement percentages ranging from 0% to 50%. We stratify on the disagreement percentage of the instances and compare *baseline* and *observed* ecosystem-level outcomes for each subset of instances.

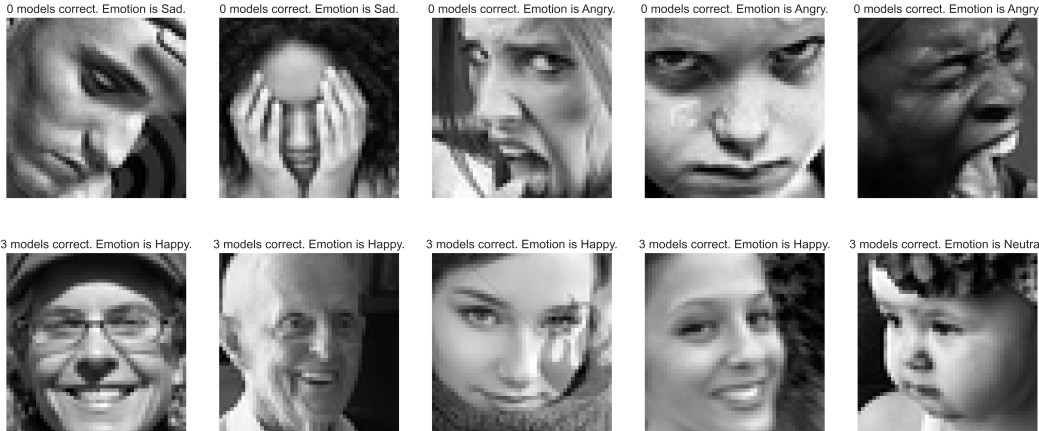

Figure 7: **Examples of homogeneous outcomes** Instances that are sampled uniformly at random from "0 models correct" (*top row*) or "3 models correct" (*bottom row*) in FER+. The systemic failures (*top row*) do not appear to be inherently harder for humans to classify; more extensive analysis appears in the supplement.

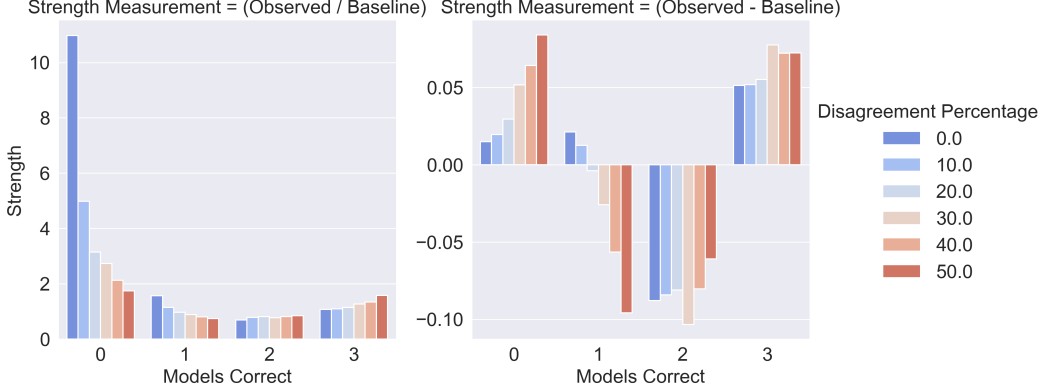

Figure 8: Interpretation of the relationship between instance-level annotator disagreement and homogeneous outcomes depends on if the strength of the effect is quantified as the *ratio* between observed and baseline rates (left plot) or as the *difference* between observed and baseline rates (right plot).

**Examples.** To build further intuition, we present several randomly sampled instances from the FER+ facial emotion recognition dataset in Figure 7.[5] We emphasize that while systemic failures may share structure, we do not believe these instances are *inherently* harder than ones on which models perform well. The authors did not have difficulty labelling these examples, nor would other human labelers. We address the question of why systemic failures arise in §6.

**Homogenous Outcomes manifest regardless of annotator disagreement.** In Figure 6, we find that homogenous outcomes appear for all levels of annotator disagreement. While more ambiguous examples exhibit higher model error rates and systemic failure rates, the observed rate of homogeneous outcomes exceeds the baseline rate of homogeneous outcomes for all instance subsets. This suggests that instance-level ambiguity does not (fully) explain the existence of homogenous outcomes— at least in FER+.

**The intensity of the effect varies by disagreement level, but the direction of the relationship depends on how strength is quantified.** In light of the observed existence of homogeneous

---

[5]We acknowledge the task of facial emotion recognition has been the subject of extensive critique [e.g. Barrett et al., 2019, Mau et al., 2021]). We provide examples for this task due to ease of visualization, but our claims also hold for examples from the text and speech modalities that are provided in the supplement.

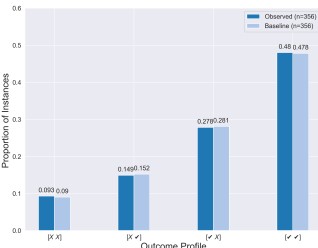 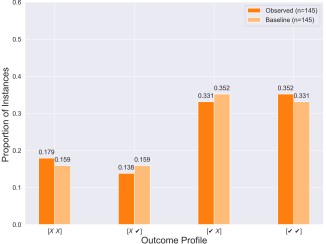 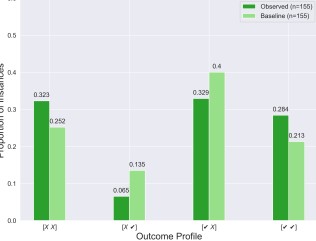

(a) Subset of instances that both dermatologists classify correctly.

(b) Subset of instances that precisely one dermatologist misclassifies.

(c) Subset of instances that both dermatologists misclassify.

Figure 9: Homogeneous outcomes exists for the subset of instances where both dermatologists misclassify the image but doesn't exist for the subset where both dermatologists correctly classify the image.

outcomes across all levels of disagreement, we now examine how the *intensity* of this effect varies with annotator disagreement. The relationship between annotator disagreement and the intensity of homogeneous outcomes depends on the quantification method used to measure homogeneous outcomes intensity. When quantifying the intensity as the difference between observed and baseline rates, the effect becomes more pronounced as disagreement increases. However, when considering the intensity as the ratio between observed and baseline rates, as in the homogenization metric introduced by Bommasani et al. [2022], the effect is most pronounced at lower levels of disagreement.

## A.2 Human accuracy

Each instance of the **DDI** dataset — a medical imaging skin lesion dataset — contains predictions from 2 models and 2 humans with the ground truth generated from an external process (in this case, a biopsy of the lesion). To understand how human-perceived difficulty relates to homogeneous outcomes, we stratify instances on the dermatologist accuracy of that instance (each instance has a dermatologist accuracy of 0%, 50%, or 100%) and examine homogeneous outcomes for each subset of instances.

**Model outcomes are homogenous for instances on which dermatologists fail and heterogeneous for on which dermatologists succeed.** In Figure 9, we find that model outcomes exhibit homogeneous outcomes for the subset of instances that both human dermatologists fail at to an even greater extent than observed at the population level. In contrast, the observed and baseline distributions match each other (meaning there is no homogeneous outcomes, heterogeneity or other form of deviation between the distributions) for the subset of instances that both dermatologists get right. Note that in the instances that both dermatologists fail at, both models systemically fail more than the baseline, but both also succeed more than the baseline. While the correlation between human systemic failures and model systemic failures seems intuitive, it's less clear why models jointly succeed more than the baseline on human systemic failures.

Given these surprising and unintuitive findings, we encourage future work to explain what features of these instances lead models to pattern together. We speculate that these instances lack sufficient informational cues, prompting the models to place excess emphasis on a narrow selection of features, consistent with the literature of machine learning models being susceptible to spurious correlations [e.g. Sagawa et al., 2020]. At face value, these results would suggest that human-level difficulty can be predictive of outcome homogeneity, but we emphasize two caveats: (i) the **DDI** dataset's sample size is limited (155 instances of dermatologists both failing and 356 instances of dermatologists both succeeding) and (ii) we rely on the judgments of just two human domain experts, meaning the findings may not generalize to larger annotator pools of non-experts (as are common in many NLP or computer vision datasets). In general, we caution against overgeneralizing this finding. We provide it as an initial foray into understanding example difficulty in a unique setting where we have data that supports an approximation of human-perceived difficulty.

## A.3 More expressive theoretical models

Finally, while we consistently find that observed ecosystem-level behaviors yield more homogeneous outcomes than the instance-level independent model predictions would predict, we might intuit there exists instance-level structure that should be encoded into the prior. Therefore, we consider a theoretical framework to encode richer priors on what we might expect of models. As a simple model, we will assume some instances are universally 'hard', meaning all models will perform worse on average across these instances, and others are conversely 'easy', meaning all models will perform better on average across these instances.

As before, we note that there is not a universal standard according to which examples can be considered 'hard.' Some examples are easy for humans but hard for models; other examples are easy for models but challenging for humans; and still other examples are challenging for some models but not others. The three hypotheses we consider in this section explore these observer-relative dimensions of hardness.

We use two parameters $(\alpha, \Delta)$ to parameterize this model, thereby adjusting the baseline rate that we calculate for ecosystem-level outcomes. $\alpha$ specifies the composition of 'hard' vs 'easy' instances in a dataset and $\Delta$ controls how much harder or easier the hard or easy examples, respectively, are expected to be. Concretely, $\alpha$ fraction of instances are categorized as difficult and the remaining $1 - \alpha$ are easy. A model's failure rate $\bar{f}_j$ over all examples scales to $\bar{f}_j^{\text{hard}} = (1 + \Delta)\bar{f}_j$ on hard examples and $\bar{f}_j^{\text{easy}} = \left(1 - \frac{\alpha\Delta}{1-\alpha}\right)\bar{f}_j$ on easy examples.

The distribution of the *baseline* number of model failures $t \in \{0, \ldots, k\}$ follows a weighted sum of two Poisson-Binomial distributions parameterized by the scaled hard $\bar{f}_j^{\text{hard}}$ and easy $\bar{f}_j^{\text{easy}}$ model error rates.

$$P_{\text{baseline}}^{\text{hard}}(t \text{ failures}) = (\alpha)\text{Poisson-Binomial}(\bar{f}_1^{\text{hard}}, \ldots, \bar{f}_k^{\text{hard}})[t] \tag{3}$$

$$P_{\text{baseline}}^{\text{easy}}(t \text{ failures}) = (1 - \alpha)\text{Poisson-Binomial}(\bar{f}_1^{\text{easy}}, \ldots, \bar{f}_k^{\text{easy}})[t] \tag{4}$$

$$P_{\text{baseline}}(t \text{ failures}) = P_{\text{baseline}}^{\text{hard}}[t] + P_{\text{baseline}}^{\text{easy}}[t] \tag{5}$$

**Identifying $\alpha$ and $\Delta$ values that recover the observed ecosystem-level outcomes in HAPI.** We utilize this framework to identify which $(\alpha, \Delta)$ combinations generate baseline distributions that recover the observed distributions in the **HAPI** datasets. We perform a grid search for $\alpha \in [0.1, 0.5]$ with a step size of 0.1 and $\Delta \in [0.2, 5]$ with a step size of 0.2. Note that certain $(\alpha, \Delta)$ combinations can result in invalid error rates depending on the original error rates of models – i.e. when $\left(1 - \frac{\alpha\Delta}{1-\alpha}\right)\bar{f}_j < 0$.

For each dataset, we identify the $(\alpha, \Delta)$ combination that minimizes the L1 distance (equivalently the total variation distance) between the observed and baseline distributions. In Figure 10, we visualize the observed and baseline distributions for the distance-minimizing $(\alpha, \Delta)$ pair for each dataset, and in Figure 11 we plot the L1 distance for each dataset as a function of $\alpha$ and $\Delta$. The majority of the best $\alpha$ values are 0.2 or 0.3 while the $\Delta$ values can range from 1 to 4 – indicating that the $\bar{f}_j^{\text{hard}}$ can be up to 5x higher than $\bar{f}_j$.

**High $\Delta$ and low $\alpha$ represents a small group of very difficult instances that all models struggle at.** The combination of high $\Delta$ and low $\alpha$ values performing well suggests that we would need to expect that a relatively small fraction of the dataset contains instances that are significantly harder for all models to perform well at. Note that this framework assumes that all models consider the same instances to be 'hard': this agreement is a form of homogenization that could be caused about something inherent to the data or something about how the models are constructed.

## B HAPI Experiments

In the main paper, we work extensively with the **HAPI** dataset of Chen et al. [2022a]. While we defer extensive details about the dataset to their work, we include additional relevant details here as well as any relevant decisions we made in using the **HAPI** dataset.

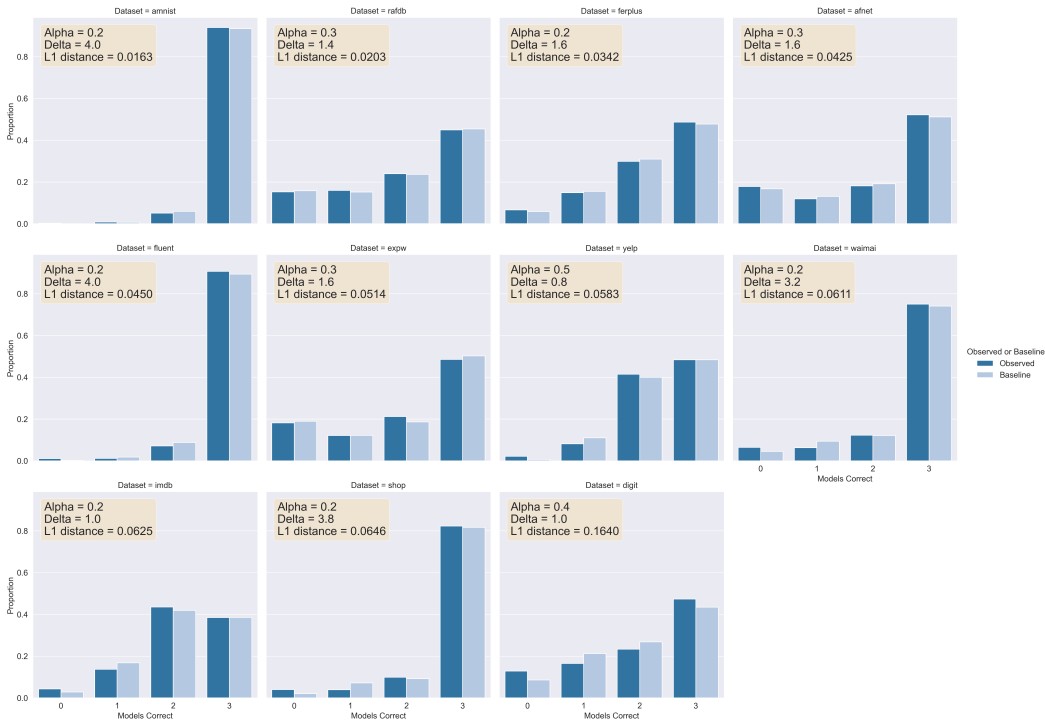

Figure 10: Observed and baseline distributions of ecosystem-level outcomes for the $(\alpha, \Delta)$ combination that yields the lowest L1 distance for each dataset.

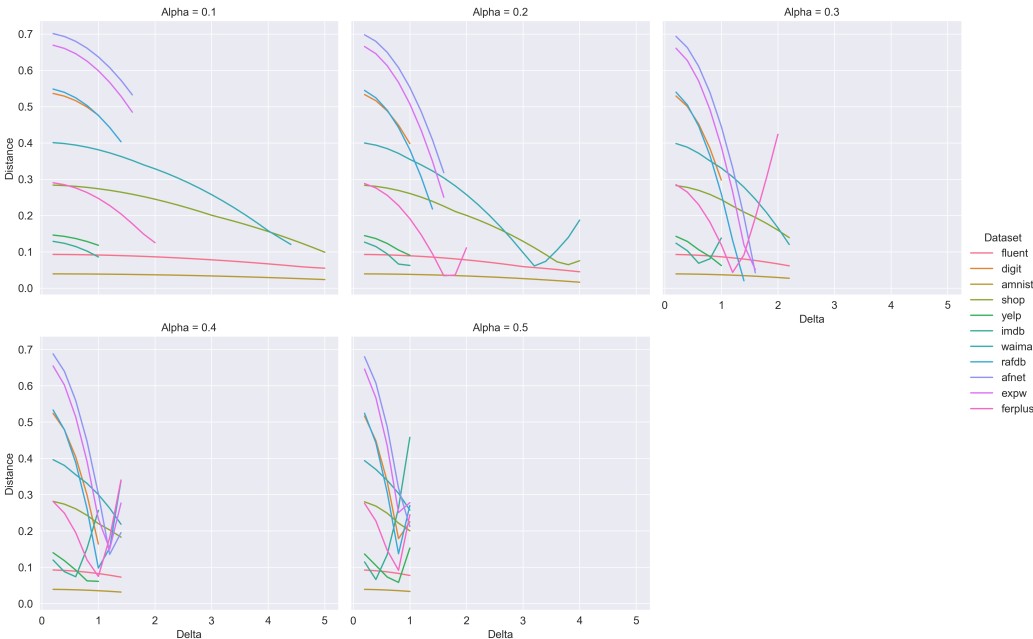

Figure 11: L1 distance between observed and baseline distributions for all datasets as a function of $\alpha$ and $\Delta$.

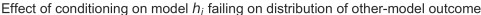

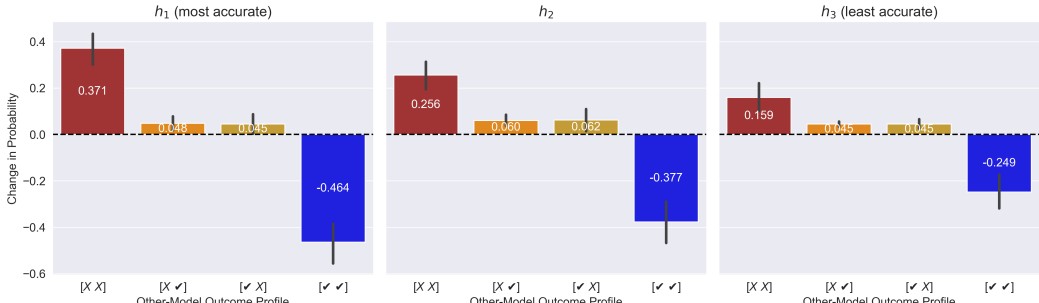

Figure 12: Change in the probability of observing each outcome profile for the other models upon observing $h_i$ fail.

## B.1 Data

We work with a subset of **HAPI**, a dataset introduced by Chen et al. [2022a] which contains predictions from commercial ML APIs on a variety of standard benchmark datasets from 2020-2022. **HAPI** contains benchmark datasets for three classification tasks and three structured prediction tasks. We only work with the classification tasks – spoken command recognition (SCR), sentiment analysis (SA), and facial emotion recognition (FER) – because they contain a single ground truth label where the notion of a 'failure' is clear (i.e. a misclassification).

The three classification tasks span 3 modalities (text, images, speech), and each task is associated with 4 datasets. However, we exclude one of these datasets: the COMMAND dataset has duplicate example IDs with differing predictions from the same ML API provider in the same time period. This makes calculating ecosystem-level outcomes impossible. After excluding the COMMAND dataset, there are 11 datasets that we conduct ecosystem-level analysis on: RAFDB [Li et al., 2017], AFNET [Mollahosseini et al., 2019], EXPW [Zhang et al., 2016], FER+ [Barsoum et al., 2016], FLUENT [Lugosch et al., 2019], DIGIT [Jackson et al., 2018], AMNIST [Becker et al., 2018], SHOP, [6] YELP, [7], IMDB [Maas et al., 2011] and WAIMAI. [8]

Each dataset contains predictions from 3 commercial ML APIs in 2020, 2021, and 2022; however the **HAPI** API did not return predictions from the Face++ model on AFNET in 2022, so we use 2021 predictions for AFNET when conducting experiments that only use predictions from a single year.

The **HAPI** dataset is distributed at `https://github.com/lchen001/HAPI` under Apache License 2.0.

## B.2 Leader Following Effects in Systemic Failure

One consequence of homogeneous outcomes is that it concentrates failures on the same users, so a user who interacts with a model and experiences a failure from that model is now more likely to experience a failure from every other model in the ecosystem. To quantify the strength of this effect, we examine how the probability of a user experiencing each outcome profile changes after that user experiences one failure from a model.

In Figure 12, we find that, consistent with the observed homogeneous outcomes in **HAPI**, observing a single model failure significantly increases the probability that the user will now experience failures from all other models in the ecosystem. However, we also find that the strength of this effect is strongly graded by the accuracy of the model for which we initially observe a failure. Upon observing the most accurate model in the ecosystem fail for a user, the probability of that user experiencing a systemic failure increases by 37% – whereas it only increases by around 16% when we observe the least accurate model fail.

---

[6]`https://github.com/SophonPlus/ChineseNlpCorpus/tree/master/datasets/online_shopping_10_cats`

[7]`https://www.kaggle.com/datasets/yelp-dataset/yelp-dataset`

[8]`https://github.com/SophonPlus/ChineseNlpCorpus/tree/master/datasets/waimai_10k`

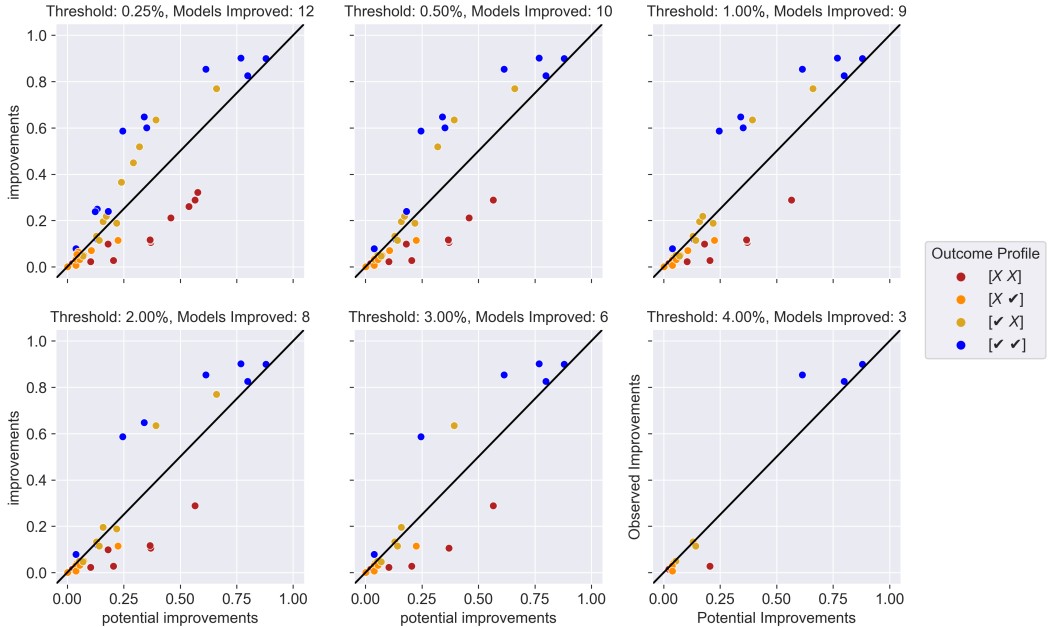

Figure 13: We replicate the graph from Figure 3 but with various thresholds for how much a model must improve for us to include it in the analysis. The patterns discussed in §4 are consistent across choice of threshold.

This result suggests that instances that the most accurate model fails on are likely to be failed by the less accurate models as well. This 'leader following' phenomenon has implications for users in a model ecosystem: users failed by the most accurate model likely have few options for alternative models that could work for them.

### B.3 Model improvement analysis is insensitive to threshold

In §4, we study how the improvement of a single model, in the sense that it becomes more impact, manifests at the ecosystem-level. To define improvement, we set the (slightly arbitrary) threshold that the model's accuracy improve by $0.5\%$, which we found to be large enough to be material to a model's performance while small enough to capture most model improvements. In Figure 13, we plot the outcome profile distribution in the observed improvements set against the distribution over the potential improvements (as in Figure 3b) for 6 different thresholds of change. Across all thresholds, the patterns we discuss in §4 hold: namely, models consistently under-improve on systemic failures. This confirms that the findings and qualitative understanding we present is not particularly sensitive to the exact value of this threshold.

### B.4 Net Improvements

In §4, we define *improvements* as the specific instances that $h_{\mathrm{imp}}$ misclassifies in the first year and correctly classifies in the second year. However, the size of this set — hereafter, *gross improvements* — is always larger than the number of *net improvements* the model makes because updates to the model tend to improve on some instances and regress on others. That is, there are two competing forces when models change: the instances the model flips from incorrect to correct, but also the instances the models from correct to incorrect. The difference of (i) and (ii) is the number of net improvements.

In Figure 14, we use net improvements instead of gross improvements, replicating the plot from in Figure 3. To calculate the outcome profile distribution over net improvements, we subtract the number of gross declines from the number of gross improvements for each outcome profile: the denominator is the number of gross improvements minus the number of gross declines across all outcome profiles.

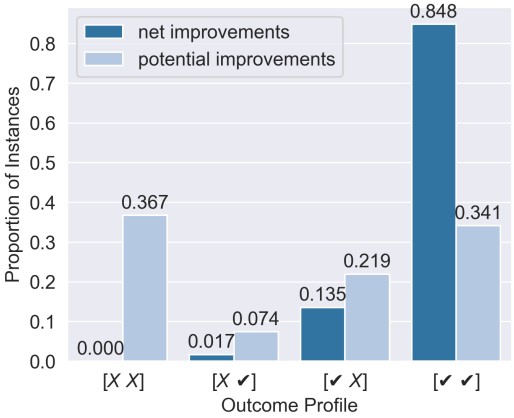 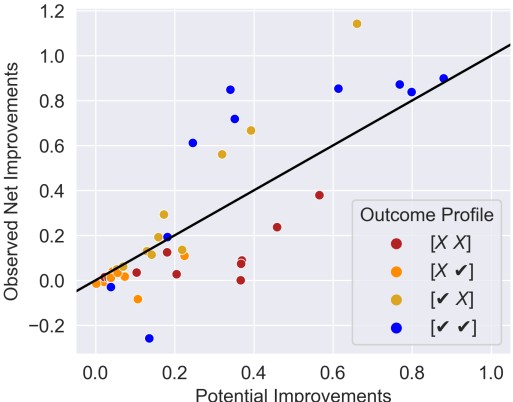

(a) Outcome profile distribution using *net improvements* instead of *gross improvements* for (Baidu, Google) when Amazon improves on WAIMAI from 2020 to 2021.

(b) The distribution of outcome profiles for *net improvements* instead of *gross improvements* for all year-over-year model improvements across all datasets.

Figure 14: We replicate the graphs in Figure 3 but using 'net improvements' instead of 'gross improvements'. The overall trends are consistent between the two experiments: namely, $h_{imp}$ under-improves on systemic failures.

The trends are generally consistent with what we observed when using gross improvements: $h_{imp}$ tends to make little progress on systemic failures. In fact, when considering net improvements, models make even less improvement on systemic failures than before. We highlight WAIMAI as a striking case study: there is no net improvements on systemic failures, despite a 2.5% decrease in model error at the population level.

Overall, we emphasize that we recommend future research conducts analyses with both notions of improvements. While we expect in many cases, as we have seen here, that the qualitative trends will be similar, the interpretations may differ. For example, gross improvements more directly attend to the concern that there are some individuals who, year-over-year, continue to be failed by some or all models in the system. In contrast, net improvements more directly matches the sense in which models are improving.

### B.5 When Models Get Worse

As a related question to what we examine in Figure 3b, we examine what happens to ecosystem-level outcomes when a model gets worse. We find that, when models get worse, they disproportionately introduce new systemic failures into the system by 'over-declining' on instances that other models were already failing for. This further highlights how single-model measurements often fails to align with ecosystem-level outcomes.

## C  Dermatology Experiments

### C.1  Data

We work with **DDI** (Diverse Dermatology Images), a dataset introduced by Daneshjou et al. [2022], which contains predictions from 3 models and 2 board-certified dermatologists on 656 skin lesion images; the task is to predict whether a lesion is malignant or benign. The ground truth label comes from an external-source: in this case, a biopsy of the lesion, which is considered the gold-standard labeling procedure in this domain.

The 3 evaluated models include ModelDerm [Han et al., 2020], a publicly available ML API, and two models from the academic literature – DeepDerm [Esteva et al., 2017] and HAM10k [Tschandl et al., 2018] – that were chosen by Daneshjou et al. [2022] on the basis of their "popularity, availability, and previous demonstrations of state-of-the-art performance." Note that, in this case, none of the models have been trained on any portion of the **DDI** dataset; the entire dataset serves as a test set.

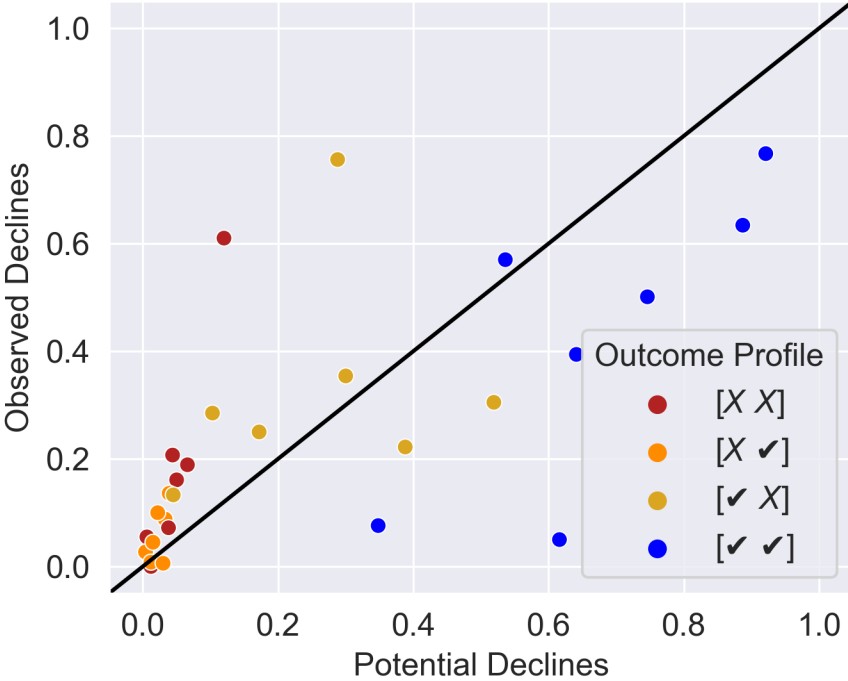

Figure 15: The distribution of outcome profiles for all year-over- year model declines across all datasets

In addition, each image is annotated with skintone metadata using the Fitzpatrick scale according to one of three categories: Fitzpatrick I & II (light skin), Fitzpatrick III & IV (medium skin), and Fitzpatrick V & VI (dark skin). For all instances, the Fitzpatrick classification was determined using consensus review of two board-certified dermatologists. Additionally, a separate group of dermatologists rated the image quality of each image and discarded any low quality images; there was no significant difference in image quality ratings between images of different FST classifications.

Data on model and dermatologist predictions was graciously provided by Daneshjou et al. [2022], subject to the terms of their standard research use agreement described in `https://ddi-dataset.github.io/index.html#access`.

## C.2 Analyses are insensitive to including/excluding HAM10k

In §5, we don't include predictions from Ham10K because the model predicts a negative on almost all instances: it has a precision of $0.99$ but a recall of $0.06$.

We decided to remove HAM10k because the pattern of near-universal negative predictions does not reflect model behavior we would expect of models deployed in clinical settings. Namely, even if deployed, the structure of the model errors are not particularly interesting and are largely predictable (in direction). Beyond these fundamental reasons for excluding the model, we also removed the model for reasons unique to our analysis. Including the model would have introduced an explicit class correlation in systemic failures (i.e. almost all systemic failures would be malignant instances and none would be benign instances) and would have complicated the comparison with humans since there would be three models but only two humans.

To confirm that our findings hold independent of this choice, in Figure 16, we replicate Figure 4 and Figure 5 but with outcomes from HAM10k included. We find that inclusion of HAM10k exacerbates the homogeneous outcomes of model outcomes and the racial disparities in model outcomes.

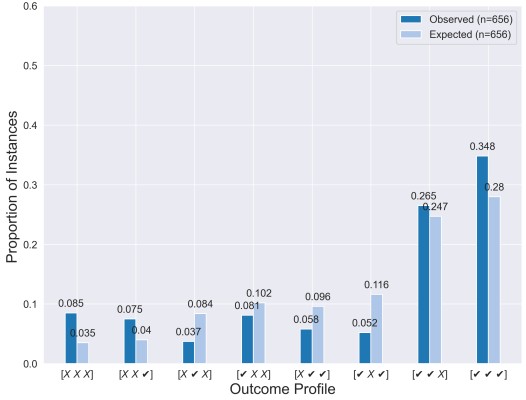

(a) Model outcomes when including HAM10K yield even more homogenous outcomes on **DDI** than in Figure 4

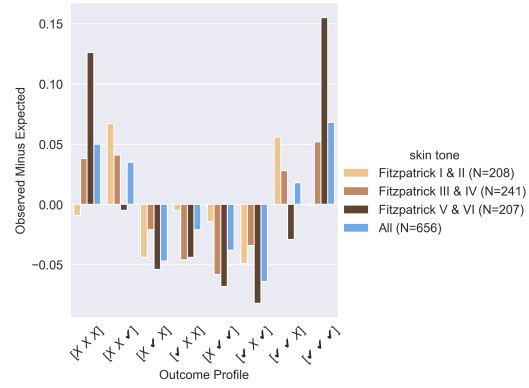

(b) The inclusion of HAM10K yields more pronounced racial disparities than in Figure 5 .

Figure 16: We replicate the graphs in Figure 4 and Figure 5 but with the inclusion of HAM10K. Homogeneous outcomes and racial disparities in models are even more pronounced when including HAM10k.

