# OpenReview forum: "Ecosystem-level Analysis of Deployed Machine Learning Reveals Homogeneous Outcomes"
_NeurIPS.cc/2023/Conference — NeurIPS 2023 poster_

### Official Review · Reviewer_KctL · 2023-07-04

**Soundness:** 3 good
**Presentation:** 4 excellent
**Contribution:** 4 excellent
**Rating:** 8
**Confidence:** 4

**Summary:**

This paper studies how particular examples in different benchmark datasets perform across multiple models using the HAPI and DDI datasets. It finds first that many examples are “systemic failures,” meaning that they are misclassified by all of the commercial models evaluated. Then, it also finds that model improvements over time do not tend to improve the rate of systemic failures. Even though some examples are no longer failing on all models, other examples can become systemic failure and there is overall nearly no net change. Finally, in the DDI dataset, the authors find differing rates of error by skin tone in human annotations, but show that the model does not have the same disparity. Overall, the paper is one of the first to study performance from this cross-model, cross-task “ecosystem-level” view.


**Strengths:**

This paper’s primary strengths are its originality, quality, and clarity. First, this paper is an original addition to the emerging homogeneous outcome literature. To my knowledge, there has yet to be a study that understands whether individual model improvements actually improve outcomes for people for whom other models have failed. The finding here that they do not is a valuable contribution to the field and important supporting evidence for the dangers of algorithmic monoculture. Second, the authors do an excellent job of clearly defining the technical limitations and strengths of their work, in particular in section 6. I found no major technical errors, and the authors make great use of the two datasets. For example, this seems like exactly the kind of study that HAPI was collected for. Finally, the paper is exceptionally clearly written.


**Weaknesses:**

This paper’s primary weakness, in my view, is specifically the model put forth in section 3.2 to compute the expected rates of failure. Though the authors do a good job in section 6 of addressing some of these concerns, I am not sure that assuming independence of the ML models is the correct probability model for this case. Because each model is tested on the same dataset, their failures by definition cannot be independent of one another. The authors’ test in section 6 with a subsample of harder examples and subsample of easier examples does offer an alternative failure model, but I’m not sure that fully captures the correlation of having the same example passed through each of the models. At the same time, I don’t necessarily feel that an “expected rate” of failure is really essential for the overall message of this paper. One option would be to remove section 3 and the corresponding results (or at least the part about expected failure rates). Another would be to just replace the model in section 3 with the slightly more detailed model of section 6, since that is at least capturing some of the correlation. I still feel that some more treatment of correlation of model performance across individual instances would be necessary though. I also recognize that this correlation is really what the paper is trying to measure empirically, which is again why I feel that a computation of expected failures is unnecessary, and most models of that expected rate are inherently unrealistic.


**Questions:**

When the paper refers to “negative outcomes”, is it referring to a misclassification or, more generally, a harmful event to a person (like being rejected for a job)? In the introduction, it wasn’t clear to me and it might be worth explicitly defining what it is for the purposes of this paper.

In section 4, could you also include information about successes that turn into failures? For example, you observe that examples that are improved on are usually ones that were already passing for another model. Is there a similar effect that successes that become failures are also examples that were already failing for another model?

In lines 242 to 243, there is a statement that systemic failure rate is monotonically correlated with annotator disagreement. I think this is a significant point towards potentially explaining the nature of systemic failures. Would it be possible to include a figure or more details on this result?


**Limitations:**

The authors do a great job of acknowledging their limitations in section 6. My only comment is that some of that material might work better at the ends of the specific sections they address. There were a couple of cases where I found myself with questions at the end of a results section that ended up getting answered in section 6.

---

> ### Author Rebuttal · Authors · 2023-08-09
>
> Thank you to reviewer KctL for a very thorough and insightful review. First, we will address the issue of the expected failure rates and discuss what we hope this rate captures and what it does not; then we answer the specific questions raised by the reviewer.
>
> > Though the authors do a good job in section 6 of addressing some of these concerns, I am not sure that assuming independence of the ML models is the correct probability model for this case
>
> To clarify, the purpose of these ‘expected failure rates’ is not to make an accurate ex-ante prediction of what the distribution of ecosystem-level outcomes will look like. Instead, we posit the baseline of independent models as a natural way to account for the error rates of the models in each ecosystem. Because the rate of systemic failure is correlated to the error rates of the model in an ecosystem (i.e. less accurate models will fail more often and thereby systemic failures will occur more often), it can be difficult to compare the rate of systemic failure across ecosystems with varying model accuracies. Additionally, it can be difficult to get a sense of what a ‘large’ rate of systemic failure is since this will depend on the error rates of the model in the ecosystem. The independence assumption is a helpful baseline because it makes no assumptions about the correlation between models in the ecosystem (which, in a sense, is what we hope to measure in our analysis), but enables us to adjust for the model error rates in the system. This is exactly what is proposed by [Bommasani et al. 2022](https://arxiv.org/pdf/2211.13972.pdf) in their formalization of outcome homogenization. We are very open to suggestions on how to better frame this baseline rate -- including a change in the name. Rather than calling this rate “expected” we could call it “baseline” or “normalized.”
>
> > When the paper refers to “negative outcomes”, is it referring to a misclassification or, more generally, a harmful event to a person (like being rejected for a job)? In the introduction, it wasn’t clear to me and it might be worth explicitly defining what it is for the purposes of this paper.
>
> This is a very good question. We intentionally use the broad term of ‘negative outcomes’ because the definition of a failure can vary by ecosystem. In our setting, we are studying classification ML models, so we concretely define the negative outcome in our ecosystem as a misclassification. However, as the reviewer points out, the term is flexible to the ecosystem; in the job hiring setting, we could define a ‘negative outcome’ as a job rejection (especially because there may be "ground truth" of whether an individual should or should not be hired); in a LM setting, we might define it as a model hallucination or misinformation produced. Once we are allowed to edit the paper, we can clarify this point in the introduction.
>
> > In section 4, could you also include information about successes that turn into failures? For example, you observe that examples that are improved on are usually ones that were already passing for another model. Is there a similar effect that successes that become failures are also examples that were already failing for another model?
>
> Another very good question; we ran this experiment, but we had to cut it from the paper due to page constraints. The figure can be viewed at the anonymous link: https://imgur.com/a/qsE3hbH.
> Our analysis confirms the reviewer’s hypothesis: when models get worse, they tend to get worse at examples that other models were already failing at. At https://imgur.com/a/qsE3hbH, we’ve attached the same graph as figure 4b, expect it displays “Observed Declines” on the y-axis and “Potential Declines” on the x-axis (in place of “Observed Improvements” and “Potential Improvements”, respectively); it does so only for periods where there is a net decline in the accuracy of a model in the ecosystem. This graph suggests that models tend to disproportionately decline on instances that other models in the ecosystem were already failing at.
>
> >In lines 242 to 243, there is a statement that systemic failure rate is monotonically correlated with annotator disagreement. I think this is a significant point towards potentially explaining the nature of systemic failures. Would it be possible to include a figure or more details on this result?
>
> For a further discussion of this result and figures on this result, please refer to Appendix section A.1. It is true that annotator disagreement is monotonically correlated with the systemic failure rate, but annotator disagreement is also monotonically correlated with the error rate of the models. Regardless of the level of annotator disagreement, the observed systemic failure rate still exceeds the independence baseline systemic failure rate – suggesting that model failures still pattern together. However, when only considering the ‘topline’ systemic failure rate and not the context of the error rates of the models, it is true that annotator disagreement can strongly predict the degree of systemic failure observed in the ecosystem.

---

> > ### Comment · Reviewer_KctL · 2023-08-16
> >
> > Thank you to the authors for their detailed response! Regarding the failure rate, I agree that a name change might clarify some of my initial confusion there - I personally like “baseline” as an alternative but leave it up to your discretion
> >
> > Thanks also for sharing the extra data on successes. I would suggest putting this into an appendix.
> >
> > After reading the response and the other reviews/responses, I am updating my score from a 7 to and 8, as I strongly believe this paper should be accepted.

---

> > > ### Author Response · Authors · 2023-08-20
> > > **Thanks for your response**
> > >
> > > Thanks for responding to our paper and your strong support backing this paper's acceptance.
> > >
> > > We will definitely include the successes results in the appendix, and will continue to think about what the best choice of term should be, but are currently also inclined towards "baseline".

---

### Official Review · Reviewer_HXE5 · 2023-07-07

**Soundness:** 3 good
**Presentation:** 4 excellent
**Contribution:** 3 good
**Rating:** 6
**Confidence:** 3

**Summary:**

The paper empirically investigates ecosystems consisting of many machine learning models. Rather than analyze each model in isolation, the paper instead focuses on the extent to which all models in an ecosystem incorrectly classify an individual (the systemic failure rate), building upon recent work on *algorithmic monoculture* (Kleinberg and Raghavan, 2021) and *outcome homogenization* (Bommasani et al., 2022).

The paper draws upon a large-scale audit (HAPI, Chen et al. 2022) that consists of 3 commercial systems per modality and 3 different modalities. They show that model improvements do not significantly reduce systemic failures: in particular, model improvements often occur on instances that are already classified by the other 2 systems. The paper then investigates two ML models (ModelDerm and DeepDerm) and two dermatologists on the dermatology dataset (DDI, Daneshjou et al., 2022). They show that the humans are more polarized (i.e. humans all agree or humans all disagree) than models. However, models (unlike humans) are more polarized for darkest skin tones than for lighter skin tones.

The paper concludes by proposing several data-centric and model-centric explanations for the observed trends.


**Strengths:**

The paper provides an interesting empirical investigation into how interactions between models within an ecosystem can affect outcomes for individuals. The results are a compelling extension of previous work on *algorithmic monoculture* (Kleinberg and Raghavan, 2021) and *outcome homogenization* (Bommasani et al., 2022).

Systemic failures is a natural metric to study, and the empirical analysis of systemic failures in model ecosystems is rich and fairly thorough. Altogether, the analysis of systemic failures on the HAPI dataset and DDI datasets provide a fairly convincing argument that a model ecosystem can exhibit different behavior a single model in isolation.

The paper is well-written and the key ideas are clearly presented.

**Weaknesses:**

There seems to be a gap between the results in Finding 4 and the implications for fairness. While Figure 6 focuses on polarization, it seems that the metric of interest for fairness should be systemic failures. This raises the question of how the systematic failure rates compare for darker skin tones and lighter skin tones. See question below.

The commentary section (Section 6) is somewhat high-level and does not provide significant insight into which factors may be driving systemic failures in a given ecosystem. As a result, it is not entirely clear what factors drive polarization and whether one should expect to see polarization in different contexts. Understanding what types of ecosystems are more susceptible to systemic failures seems like an important question for policymakers and researchers. See question below.

**Questions:**

Questions:
- How do the systemic failure rates compare for darker skin tones and lighter skin tones for the models? What about for the humans? How much larger is the gap between systemic failures that the gap between failure rates for these two groups on an individual model/human?
- Please comment on which of the factors in Section 6 are likely to most responsible for systemic failures in different contexts. What factors of a given ecosystem might influence the level of systemic failure observed?



**Limitations:**

The authors adequately addressed the limitations of their work.

---

> ### Author Rebuttal · Authors · 2023-08-09
>
> Thank you to reviewer HXE5 for a thoughtful review and set of questions. At the heart of the matter, we see two core concerns raised in the review:
> 1. **Finding 4.** We agree that systemic failures are definitely a phenomenon of direct interest, but even beyond systemic failures, the racial disparities in polarization pose substantive concerns. We provide a longer-form discussion below that we had to abridge in the paper due to space, and would be keen to include this discussion with the additional page granted upon acceptance.
> 2. **Discussion in Section 6.** This is an oversight on our part: every paragraph in Section 6 corresponds to a detailed analysis presented in the Appendix. We encourage the reader to read Appendix A, as we believe it has all of the desired rigor in these analyses, and have replicated some of the points below.
> Overall, we hope our responses are clear and convincing answers to the questions posed.
>
> > There seems to be a gap between the results in Finding 4 and the implications for fairness. While Figure 6 focuses on polarization, it seems that the metric of interest for fairness should be systemic failures. This raises the question of how the systematic failure rates compare for darker skin tones and lighter skin tones. See question below.
>
> Systemic failure rate is certainly of direct interest to fairness: we do not intend for this point to be undermined by our discussion of finding 4. However, we believe there are many important contexts where it is not enough. Namely, we believe the racial disparities in polarization warrant interest, even if they do manifest in racial disparities in the systemic failure rate.
>
> We focus on *polarization* instead of the systemic failure rate because the systemic failure rate is partially driven by the error rate of the models being considered. If all of the models under consideration have high error rates, systemic failures would occur just because of the collisions of those errors. We agree that outcomes are ethically important, no matter how they are produced. However, we hope to distinguish systemic failure driven by very high error rates from systemic failure driven by similarity between models.  The first phenomenon is consequential but has been extensively studied in literature on fairness metrics; the second phenomenon is part of the contribution of this paper.
>
> > How do the systemic failure rates compare for darker skin tones and lighter skin tones for the models? What about for the humans? How much larger is the gap between systemic failures that the gap between failure rates for these two groups on an individual model/human?
>
> Thank you for the opportunity to elaborate on this point. In the DDI data, the observed systemic failure rate for models is .045 higher for dark skin tones than light skin tones. However, this difference in even more dramatic when considering that the expected number of systemic failures for dark skin is actually *lower* than the expected number of systemic failures for light skin -- a result of the models having lower error rates for dark skin than light skin.  Considering the systemic failure rate without additionally considering the error rates would obscure this disparity.
>
> We note that the original study used ROC-AUC with the models’ outputted probabilities to measure disparities across race groups and found that the models’ ROC-AUC was higher for light skin tones than dark skin tones.  By contrast, we are considering classification error and, due to the class imbalance in this dataset, we observe a different pattern.
>
> In humans, the systemic failure rate is actually 0.023 lower for dark skin than light skin; however, the expected failure rate is also 0.024 lower -- nearly equivalent -- due to the lower error rate for dark skin than light skin.
>
> At a high level, we believe that polarization is an additional dimension of fairness that stakeholders should consider. In this one setting, we’ve seen that three different metrics paint very different pictures: using ROC-AUC suggests that both humans and models perform *worse* for dark skin than light skin; using classification error suggests that both humans and models perform *better* for dark skin than light skin; and using polarization suggests that models perform *worse* for dark skin than light skin while humans don’t exhibit any racial disparity. Stakeholders already consider various forms of fairness when evaluating decision makers -- automated or not -- and we hope that profile polarization can add one additional dimension to existing measures of fairness.
>
> > Please comment on which of the factors in Section 6 are likely to [be] most responsible for systemic failures in different contexts. What factors of a given ecosystem might influence the level of systemic failure observed?
> Appendix A presents three initial explorations of different data-centric explanations for systemic failures. These explanations complement the several prior works that are cited, which pose model-centric explanations (e.g. the sharing of model components in [Bommasani et al. 2022](https://arxiv.org/pdf/2211.13972.pdf)). Unfortunately, due to the absolute opacity of these deployed ML APIs, we have no real means for testing model-centric explanations at the moment.
> More deeply, for data-centric explanations, we explore how annotator disagreement in the ground truth label impacts homogenous outcomes, how human error correlates with outcome profiles in models, and how more expressive models that account for different levels of difficulty in input instances can better recover the observed distribution of ecosystem-level outcomes.
>
> In practice, determining what is “most responsible” will be challenging as with any multi-faceted causal analysis: we suspect that both data-centric explanation and model-centric explanations play a role in homogenous outcomes, and we encourage future work to clarify these relationships.

---

> > ### Comment · Reviewer_HXE5 · 2023-08-16
> >
> > Thanks to the authors for their detailed response! I am satisfied with the response and am maintaining my score (weak accept).

---

> > > ### Author Response · Authors · 2023-08-20
> > > **Thanks for your response**
> > >
> > > Thanks for responding to and engaging with our rebuttal, we appreciate it.

---

### Official Review · Reviewer_N5w8 · 2023-07-07

**Soundness:** 3 good
**Presentation:** 4 excellent
**Contribution:** 2 fair
**Rating:** 5
**Confidence:** 3

**Summary:**

The paper introduces ecosystem-level analysis where instead of a single model, a collection of models will be evaluated together and their common failures/successes will be identified. Across three modalities, the paper shows a pattern of polarization where an instance is either more likely to be misclassified by all the models or correctly classified by all of them. Similar patterns are observed in medical imaging. The paper suggests potential mechanisms responsible for polarization and concludes with a discussion of implications for researchers and practitioners.

**Strengths:**

- The framework appears promising to me because it centers people in the analysis and highlights how repeated failures of machine learning models for an individual can result in harm.
- It provides concrete findings for previously mostly conceptualized homogenous outcomes.
- It discovers a new form of racial parity that couldn't be identified without ecosystem analysis and is not present in human judgments.

**Weaknesses:**

1. It could be more interesting if systematic failures were studied by linking modalities rather than each modality in isolation. But I understand the available dataset does not support this.
2. As discussed in Sec 6.1, the baseline of independent models is very unrealistic, for example, difficulty may confound the outcomes. In the current baseline, the majority vote of different models significantly improves their accuracy which is not at all expected.
3. Sometimes it's not clear what level of polarization is significant. For example, Fig 5a looks as no polarization to me.
4. I was expecting more models to make a conclusion about polarization. In many places, only two models are used to draw conclusions.
5. Since the number of models is always small, I have no good sense of how the effect scales with more models.
6. To address the last two points, authors may want to increase the number of models by evaluating some models on the same datasets themselves. This will be also helpful in experimenting some conjectures about the polarization mechanisms.
7. The new form of disparity (Finding 4) is interesting, but the framing seems to obscure the fact that humans are still more polarized on all skin tones. I'd recommend emphasizing this.
8. As a general comment, I'd like to see an expanded discussion on the causes of polarization and its implications.


Typos: line 19: remove do, table 1: $h^1 \rightarrow h_1$

**Questions:**

No specific questions. Feel free to address the weaknesses I mentioned if you think they are not the case.

---

> ### Author Rebuttal · Authors · 2023-08-09
>
> We appreciate reviewer N5w8’s thorough review, highlighting the core framework and associated findings as key strengths. The questions they pose are definitely of interest but, overall,we unfortunately cannot answer several of them due to the lack of available data. Nonetheless, we provide more specific answers when possible.
>
> > It could be more interesting if systematic failures were studied by linking modalities rather than each modality in isolation. But I understand the available dataset does not support this.
>
> We agree that cross-modal relationships would be interesting, but the HAPI dataset of Chen et al. does not track the same individuals across modalities and we are unaware of any such linked dataset auditing commercially deployed models currently exists.
>
> > As discussed in Sec 6.1, the baseline of independent models is very unrealistic, for example, difficulty may confound the outcomes.
>
> We believe there is a misunderstanding in regards to the independent baseline; we would be happy to reword the paper to make this clearer. Namely, we neither claim nor intend for the independent baseline to be seen as a realistic prior for the amount of correlation we will see in practice. Instead, we provide it as an example to contextualize and contrast the observed behavior against. In particular, it accounts for the underlying error rates of the different models.
>
> Overall, we find this baseline to be a simple and natural starting point for building intuition on a new phenomenon (i.e. the failure matrix and associated ecosystem-level outcomes). As we develop in 6.1 and the associated appendix A, we can consider many more sophisticated models of model correlation: for example, the alpha-delta model we introduce in Appendix A (and describe in section 6.1 lines 255-264) generalizes independence as a simple two-parameter model that incorporates a notion of “difficulty” as the reviewer suggests.
>
> We wonder if the issue rests on the use of the term “expected”, which we use to contrast with “observed” (as is conventional in several fields); we are very open to suggestions on how to better frame this baseline rate -- including a change in the name. Rather than calling this rate “expected” we could call it “baseline” or “normalized.”
>
> > In the current baseline, the majority vote of different models significantly improves their accuracy which is not at all expected.
>
> Could the reviewer clarify what they mean by “majority vote of the different models” so we can address this concern?
>
> > I was expecting more models to make a conclusion about polarization. In many places, only two models are used to draw conclusions.
>
> To clarify, for almost all results (i.e. all HAPI results), they are based on 3 commercial models, not two. Overall, these are the 3 most widely used APIs for their respective tasks/modalities as described in the HAPI paper of Chen et al. We do agree that in the future, more controlled analyses that understand how the phenomena scales in the number of models would be interesting. However, for our focus on deployed ML systems, we believe the setting is very realistic where a few major ML-as-a-service (MLaaS) providers dominate each of the respective markets.
>
> As a potential point of clarification, we note that the analyses for model improvements depict two models because the third model (i.e. the one that improved) is being excluded to understand how its improvements relates to the remaining two models.
>
> > Since the number of models is always small, I have no good sense of how the effect scales with more models. To address the last two points, authors may want to increase the number of models by evaluating some models on the same datasets themselves. This will be also helpful in experimenting some conjectures about the polarization mechanisms.
>
> As we note above, we agree that understanding the scaling behavior is valuable. However, our focus is on the realities of deployed ML systems in these domains: it is a market reality that in many of these domains, only a few MLaaS providers dominate the respective markets (as was also described by Chen et al. in the HAPI paper). The models we study are created by the largest ML providers and, while precise market understanding remains opaque, the evidence we have found shows that these deployed models have major impact: for example, in 2019, Microsoft -- whose API systems are included in HAPI -- [claimed](https://azure.microsoft.com/en-us/blog/companies-of-all-sizes-tackle-real-business-problems-with-azure-ai/) that its ML APIs processed 5 billion ‘cognitive search transactions’ (their term for an API call) each month. While we could have trained our own models and applied them to these datasets, we believe that the societal impact of these deployed models and the interest in understanding their risks from policymakers and the public warranted our initial focus on deployed models.
>
> We should also keep in mind that the relevance of scaling behavior will depend on the ecosystem being evaluated; in the context of job hiring, for example, scaling behavior could be very relevant since job seekers tend to apply to many jobs, while in automated speech recognition systems it might be less relevant since users are likely to only interact with systems from a limited number of companies.
>
>  > framing seems to obscure the fact that humans are still more polarized …  I'd like to see an expanded discussion on the causes of polarization and its implications. … typos
>
> Given the page constraint, we were forced to include expanded discussion on both of these topics in the appendix - see lines 511-532, 593 - 608. We will uplift these discussions, directly addressing these concerns, when afforded an additional page upon acceptance.

---

> > ### Comment · Reviewer_N5w8 · 2023-08-20
> >
> > Thanks for the clarification points.
> >
> > My main concern, similar to reviewer KctL, is with the independence assumption for the baseline and expected rate notion. I agree with the authors it is a natural choice, but I’d like to see a framing that puts less emphasis on this number; being more polarized than this baseline might be confounded with many factors and it's not a sufficient evidence for homogenization. What reviewer KctL suggested in this regard sounds good to me.
> >
> > Overall I’d like to keep my current rating. I believe a solid original empirical work well suits NeurIPS.

---

> > > ### Author Response · Authors · 2023-08-20
> > > **Thanks for your response**
> > >
> > > Thanks for responding to our rebuttal, we sincerely appreciate it. We agree that it is important to figure out a standard for what constitutes "problematic" homogeneity, and that this likely will not be just comparing to this independent baseline. In particular, we imagine this will jointly come about through greater empirical understanding of the prevalence + impact of homogeneity (to calibrate how sensitive we should be/set thresholds of importance) and the development of a parallel testing regime (to distinguish problematic vs. non-problematic homogeneity on the basis of the stated threshold). Our hope is this work can get the ball rolling on the empirical side, facilitating theoretical/statistical work by providing insight on what appropriate thresholds are for any such test.

---

### Official Review · Reviewer_kuru · 2023-07-07

**Soundness:** 3 good
**Presentation:** 3 good
**Contribution:** 3 good
**Rating:** 5
**Confidence:** 3

**Summary:**

This paper introduces the concept of "ecosystem-level analysis," which evaluates the performance and impact of a collection of machine learning models in a given context rather than analyzing individual models in isolation. This approach could significantly contribute to understanding the broader impacts of deployed machine learning models on society. The authors identify a new phenomenon called "systemic failure" in machine learning models. This happens when a user is consistently misclassified by all available models in the ecosystem, pointing towards potential systemic biases in machine learning deployments.

**Strengths:**

This paper provides empirical evidence of these systemic failures across three different modalities (text, images, speech) and 11 datasets, establishing a broader trend that transcends specific applications or domains.

**Weaknesses:**

The study under review makes an ambitious attempt to analyze the societal impact of machine learning (ML) from an ecosystem-level perspective. Although the authors should be commended for their unique approach, the paper has notable gaps that prevent its acceptance, particularly for a high-standard conference such as NeurIPS. The paper's theoretical framework lacks rigor and the theoretical grounding is insufficient, making it more suitable for a venue like FAccT, which is more focused on ethical and societal aspects of ML.

The authors propose the idea of systemic failure in deployed machine learning, which is indeed intriguing. However, the theory behind this proposition appears to be weak and not well-articulated. The paper lacks robust theoretical to support this claim, as the analysis seems to be limited in terms of the ML models considered. For example, what are the factors that could contribute to system failure? Does these factors differ in terms of features distribution and machine learning algorithm used? Furthermore, the implications of such systemic failures on society, an aspect of paramount importance, is not adequately addressed, making it difficult to comprehend the real-world significance of these findings.

The authors conducted a large-scale audit covering three modalities and 11 datasets. While this might sound impressive initially, the appropriateness of the methodology is questionable. There is a lack of clarity about the selection criteria for these modalities and datasets, and their representativeness of the ML ecosystem as a whole. This omission undermines the credibility of the results. This is again because of the lack of theoretical analysis, that may make the result questionable -- does the result only hold on the datasets at hand? Had another ML method, data distribution changed, would there still be system failure?

The claim that improvements in individual models do not significantly reduce systemic failures is an interesting observation. However, the paper fails to provide a detailed theoretical examination and mechanism analysis of this phenomenon. A stronger theoretical grounding for this observation would help the NeurIPS community to better appreciate and understand this claim.

The study on medical imaging, especially in dermatology, opens up a promising area of research. However, the introduction of a new form of racial disparity unseen in human predictions is a serious claim that requires more in-depth explanation and rigorous validation across a broader range of ML methods. Without comprehensive exploration, the validity of this claim remains dubious. I actually think that the paper can focus on this unique dataset, provide more mechanism analysis using one single dataset. And then validate the conjecture on other datasets.

**Questions:**

See weaknesses.

**Limitations:**

See weaknesses.

---

> ### Author Rebuttal · Authors · 2023-08-09
>
> Thanks to reviewer kuru07 for their helpful review. Overall, we firmly disagree with many of the vague claims made in the review, and ask the reviewer to provide specific evidence for us to be able to better engage with their claims. For example, at many points there are claims of lacking empirical rigor that are unsubstantiated: the work studies 3-4 datasets, 11 ML systems, and 12 datasets in total, so we are very surprised to hear concerns about empirical rigor.
>
> Below, we provide a more specific response to the weaknesses provided.
>
> > The paper's theoretical framework lacks rigor and the empirical evidence is insufficient, making it more suitable for a venue like FAccT, which is more focused on ethical and societal aspects of ML.
>
> Work on the ethical and societal aspects of ML is directly within scope for NeurIPS 2023, named explicitly in the call for papers as “Social and economic aspects of machine learning (e.g., fairness, …)”. In fact, the award-winning paper of Birhane et al. has provided a direct critique of NeurIPS having historically undervalued the societal dimensions of machine learning. As machine learning is deployed to millions of people, including through the very deployed systems we have studied, it is absolutely essential that work on its impact is presented and discussed at top ML venues like NeurIPS.
>
> > The authors propose the idea of systemic failure in deployed machine learning, which is indeed intriguing. However, the theory behind this proposition appears to be weak and not well-articulated. The paper lacks robust empirical evidence to support this claim, as the analysis seems to be limited in terms of the ML models considered.
>
> This paper contributes an extensive empirical analysis of deployed machine learning systems provided by major vendors (e.g. Amazon, Baidu, Google, IBM) across a range of modalities spanning 10+ datasets. We are unsure of what empirical evidence is lacking (our findings of polarized profiles, for example, hold consistently in all contexts) or in what sense the models are limited. This paper does not claim to make a theoretical contribution, instead building on the theoretical framing presented in [Bommasani et. al. 2022](https://arxiv.org/abs/2211.13972), which formalizes outcome homogenization in terms of systemic failures experienced by users interacting with multiple machine learning systems.
>
> > Furthermore, the implications of such systemic failures on society, an aspect of paramount importance, is not adequately addressed, making it difficult to comprehend the real-world significance of these findings.
>
> We strongly disagree: we do explicitly discuss the importance of these failures in several places, and point more extensively to literature that has already presented these arguments such as section IV.C.4 of Ajunwa 2021, section 6.1 of Bommasani et al. 2022, and Creel & Hellman 2022. As stated, our objective in this work is not to further argue this point, but to provide robust empirical evidence of the harms that these prior works have conjectured and conceptualized.
>
> >There is a lack of clarity about the selection criteria for these modalities and datasets, and their representativeness of the ML ecosystem as a whole. This omission undermines the credibility of the results.
>
> We strongly disagree, having provided explicit justification for both the HAPI dataset in lines 108-117 and the DDI dataset in lines 189-201.
>
> Namely, the HAPI dataset is the only dataset we know of that includes predictions from multiple deployed commercial models from the largest ML as a service (MLaaS) providers (Google, Amazon, Microsoft, IBM) on the same set of instances. According to one market research report, the ML as a service market was reported to be 21.55 billion dollars in 2022, with a compound annual growth rate of over 30 percent. The major cloud players (Google, Microsoft, and Amazon) are expected to be major players in the ML as a service market, and are all present in our dataset. Beyond this, the dataset ensures we cover 3 modalities (text, image, speech) with 10+ datasets, which amounts to our findings having substantial generalizability. Further, the dataset is longitudinal, enabling out analysis of systemic failures over time.
>
> Similarly, the DDI dataset is the only dataset we know with instance-level predictions for multiple machine learning models in dermatology (namely some of the most high-profile models in the domain) and multiple board-certified dermatologists. Additionally, as described by Daneshjou et al, the dataset is one of the most diverse in the entire dermatology area, enabling our study of race-based effects. Finally, the dataset has ground-truth labels determined on the basis of a highly reliable medical procedure rather than human-based judgment, which proves to be essential for meaningfully compared model-based and human-based outcomes.
>
> Given the unique properties of this dataset, its inclusion of the major players in the market, and the importance of the ML services provided by these providers, we believe that findings of systemic exclusion are important on their own.

---

> > ### Comment · Reviewer_kuru · 2023-08-20
> >
> > First of all, my sincere apology for the typo in my review -- I meant to say theoretical grounding instead of empirical analysis. I do agree that the authors evaluated upon a rich set of datasets (as stated in my review).  I thank the authors for the responses and for the explaining the relevance to the Neurips community, as well as clarifying that the main point for the paper is to provide empirical evidence of the harm, rather than theoretical contribution. The further explanation on the DDI data is interesting, and I do think this makes the paper stronger.
> >
> > Therefore, I've raised my rating to 5. However, because of the lack of theoretical analysis/mechanisms analysis, it is unclear what could be the root cause of system failure, I cannot further relevant the rating.

---

> > > ### Author Response · Authors · 2023-08-20
> > > **Thanks for the response to our rebuttal**
> > >
> > > Thanks for your response to our author response and we are glad we could come to greater agreement. We absolutely agree deeper theoretical grounding is important, though we think this can be fairly seen as a place for future work to address whereas this paper is more squarely empirical in nature. If accepted, we will add a clear indication that future theoretical work would be encouraged, especially now that it is clear that the phenomena conceptualized in prior work have a clear empirical basis in widely-deployed ML systems.

---

### Author Response · Authors · 2023-08-20
**Update following discussion period**

We thank all four reviewers for engaging with our rebuttals. Overall, we believe the updated scores of 5, 5, 6, 8 provide strong evidence to back accepting this paper on the basis of strong empirical work, conceptual clarity, and clear exposition/presentation. We agree with multiple reviewers that a stronger theoretical basis would be an important contribution, one that we see as appropriate for future work to address.

To summarize the changes we will make, including how we plan to use the additional page granted upon acceptance:
1. **Framing.** One area which the author response helped to clarify is confusion around our use of "expected" to describe the comparison we make to the observed phenomena, namely the ecosystem-level behavior we would observe if model failures were independent of each other. While we chose "expected" to contrast with "observed", it seems clear that a term like "baseline" would better describe what we intend.
2. **Importance of systemic failures.** Due to page limit, we abridged our discussion of the societal significance of systemic failures, largely deferring this to past works that have conceptualized the matter. With that said, with the additional space, we plan to bring more of this discussion into the main paper. Also, more specific to our datasets, we provide evidence of the market-level impacts of MLaaS in our rebuttals that we can also include if reviewers/ACs feel it is helpful context.
3. **Discussion of Finding 4 on racial disparities in medical imaging.** As currently presented, we identify and highlight the importance of a racial disparity in ecosystem-level outcomes for models that does not manifest for humans. We believe this is important, especially since it is not something currently being considered in the literature, but as Reviewer HXE5 highlights, we can do a better job of explaining why, while also highlighting the higher systemic failure rates for all races/skin tones for humans than models.
4. **Encouraging future work that strengthens the theoretical foundations.** As multiple reviewers highlight, there is room for stronger theoretical understanding, including establishing various (potentially causal) explanations for why/how homogenization arises. Having thought about this extensively ourselves, we will try to provide guidance for future work to address this important matter.

Finally, we will do additional passes over the writing and also address the typos identified by reviewers. Overall, we are glad to see the reviewers recognizing the importance of this work on improving our ability to characterize the societal impact of ML, especially in concrete and widespread deployments.

---

### Decision · Program_Chairs · 2023-09-21

**Decision:**

Accept (poster)

**Comment:**

The paper empirically investigates the systemic failure of multiple models for certain segments. This is an important phenomenon that is studied well in the paper. It makes a valuable addition to the area. A consistent concern was whether the empirical insights can be augmented with theoretical one. To the extent this can be fleshed out within the scope of the paper that would add further value.